# Detecting Scarce and Sparse Anomalous: Solving Dual Imbalance in Multi-Instance Learning

## Abstract

In real-world applications, it is highly challenging to detect anomalous samples with extremely sparse anomalies, as they are highly similar to and thus easily confused with normal samples. Moreover, the number of anomalous samples is inherently scarce. This results in a dual imbalance Multi-Instance Learning (MIL) problem, manifesting at both the macro and micro levels. To address this "needle-in-a-haystack problem", we find that MIL problem can be reformulated as a fine-grained PU learning problem. This allows us to address the imbalance issue in an unbiased manner using micro-level balancing mechanisms. To this end, we propose a novel framework, Balanced Fine-Grained Positive-Unlabeled (BFGPU)-based on rigorous theoretical foundations. Extensive experiments on both synthetic and real-world datasets demonstrate the effectiveness of BFGPU.

## 1 Introduction

In real-world applications, there is a strong demand for effective detection of anomalous samples, such as in quality inspection, risk control, and fault diagnosis (Görnitz et al., 2013; Pang et al., 2023; Ye et al., 2023). Ideally, anomalous samples exhibit significant and easily distinguishable differences from normal samples, making them readily separable based on features or representations. However, in more realistic scenarios, the differences between anomalous and normal samples are often subtle because anomalies may manifest as sparse local information. As a result, anomalous samples are highly similar to normal ones at a macro level. Moreover, the number of anomalous samples is inherently scarce in reality, making them more difficult to identify.

On the one hand, this problem has significant practical relevance. For instance, in cancer cell detection, diseased samples are inherently rare, and within these samples, the pathological regions often occupy only a minuscule portion. This makes early detection before the cancer metastasizes exceptionally challenging. On the other hand, the rise of large language models (LLMs) has further amplified the importance of this detection problem. During the Reinforcement Learning from Human Feedback (RLHF) phase, aligning LLMs with human values requires training a separate detector as a reward model (Bai et al., 2022). This model is trained on human feedback to identify non-compliant sections in the LLM's outputs. However, these non-compliant contents are both scarce and sparse. Compounding this is the high cost of human annotation, which typically only provides coarse-grained labels. These factors make training an accurate reward model extremely difficult, thereby further highlighting the critical nature of this detection challenge. An example is provided in fig. 1, a real-world customer service quality inspection task, the goal is to detect instances of impoliteness within extended conversations between customer service and customer. Since the customer service is well-trained, mistakes—if any—typically occur in only a subtle utterance.

Given the difficulty of distinguishing samples at the macro level, a promising approach is to seek solutions at the micro level—by decomposing each macro-level sample into multiple finer-grained components and performing discrimination at this finer resolution. However, there are no precise labels available at the micro level, so learning must rely solely on macro-level supervision. Multiple Instance Learning (MIL) is a paradigm of weakly supervised learning designed for scenarios with imprecise labels (Waqas et al., 2024; Zhou, 2018; Gao et al., 2022). It structures data into two hierarchical levels: the macro level (referred to as bags) and the micro level (referred to as instances), and utilizes only macro-level labels for training. To address the lack of micro-level supervision, existing MIL methods often heuristically assign bag labels to all instances within the bag (Ilse et al.,

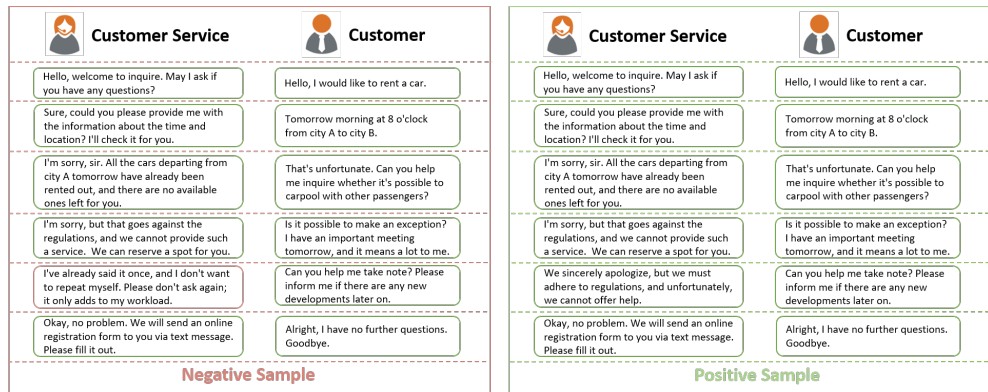

Figure 1: This example illustrates the difficulty in distinguishing between normal and anomalous macro samples due to the low proportion of anomalous information within anomalous samples.

2018; Angelidis & Lapata, 2018; Pang et al., 2023). However, this strategy lacks theoretical grounding and can introduce substantial bias. In terms of data imbalance, the micro-level imbalance—known in MIL literature as the low witness rate problem—has been recognized but remains unresolved (Carbonneau et al., 2018; Zhang et al., 2022). Meanwhile,macro-level imbalance has been largely overlooked and presents an even greater challenge.

In this paper, instead of directly assigning bag labels to instances, we treat all instances from normal bags as positive instances and all instances from anomalous bags as unlabeled instances. This reformulates the MIL problem as a fine-grained Positive-Unlabeled (PU) learning task. PU learning is a learning paradigm that trains models using only positive and unlabeled data (Bekker & Davis, 2020). Building upon this reformulation, we further derive a PU learning loss function based on rigorous theoretical analysis, which simultaneously addresses the dual imbalance challenge in MIL.

To develop a PU learning algorithm better suited for addressing MIL problems, we further incorporate macro-level information into the micro-level learning process. Specifically, we restrict the assignment of pseudo-labels to only high-confidence unlabeled instances, and dynamically adjust the confidence threshold to maintain balanced prediction. This design enables the model to improve macro-level performance directly through micro-level learning. The resulting algorithm, termed Balanced Fine-Grained Positive-Unlabeled (BFGPU), is a PU learning method tailored for solving MIL problems.

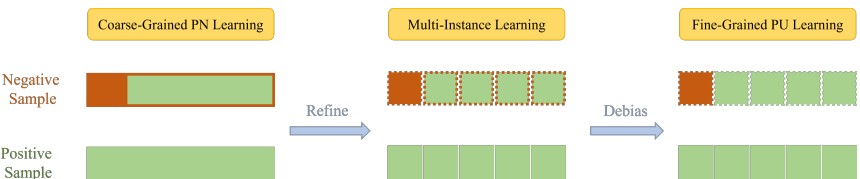

Figure 2: This figure illustrates the different solution paradigms for the detection problem in this paper: coarse-grained positive-negative learning, MIL, and fine-grained positive-unlabeled learning.

In the text and image modalities, we conducted comprehensive comparisons with existing supervised learning, anomaly detection, MIL, and PU learning methods based on customer service quality inspection (CSQI) and invasive ductal carcinoma (IDC) tasks, respectively. Furthermore, to validate the robustness of our algorithm under varying macro and micro imbalance ratios, we synthesized numerous datasets with different imbalance ratios based on four fundamental sentiment analysis datasets for additional comparative experiments. We also compared its performance with popular LLMs in addressing the "needle in a haystack" problem (Wang et al., 2024a; Kuratov et al., 2024). Finally, through ablation studies and parameter sensitivity analysis, we further enriched the discussion on the algorithm's robustness.

**Our Contributions.** 1. We formalize the challenging detection problem in practical applications and propose a dual imbalanced MIL problem. 2. We propose a novel perspective that reformulates the MIL problem as a balanced PU learning task, which addresses the issues of bias and imbalance. 3. We incorporate macro information into the learning process, leading to BFGPU, an algorithm that enables

balanced and unbiased macro performance by training at the micro level. 4. We provide a theoretical analysis demonstrating the advantages of BFGPU over macro-level learning and conventional MIL approaches. 5. Extensive experiments validate the superior performance of BFGPU.

## 2 RELATED WORKS

**PU learning** can utilize positive data and unlabeled data to train a classifier that distinguishes between positive and negative data (Bekker & Davis, 2020). Most PU learning algorithms aim for the model to be unbiased in terms of accuracy, and there are many effective algorithms currently available (Du Plessis et al., 2015; Kiryo et al., 2017; Chen et al., 2020; Kato et al., 2018; Shi et al., 2018; Sansone et al., 2018; Hou et al., 2018; Hsieh et al., 2019; 2015; Wang et al., 2024b). Some PU learning algorithms address the issue of imbalance in the positive-negative ratio within unlabeled data and strive for unbiased average accuracy (AvgAcc) (Su et al., 2021). Additionally, theoretical research on PU learning has demonstrated its superiority (Niu et al., 2016).

**Multi-instance learning (MIL)** is a form of weakly supervised learning (Zhou, 2018) characterized by inexact supervision, where labels are not precise (Carbonneau et al., 2018). In MIL, a single label is assigned to a bag of instances, addressing the problem we are discussing. In the past, MIL approaches typically involved pooling or attention mechanisms to fuse embeddings or setting up scoring functions (Ilse et al., 2018; Feng & Zhou, 2017; Pinheiro & Collobert, 2015; Perini et al., 2023; Pang et al., 2023; Abati et al., 2019). In our case, we aim to transform this problem into a direct micro PU learning problem to achieve more accurate solutions and eliminate redundant information.

**Anomaly Detection (AD)** the process of detecting data instances that significantly deviate from the majority of data instances (Pang et al., 2021). Historically, AD methods have been categorized into supervised AD (Görnitz et al., 2013), weakly supervised AD (Ruff et al., 2019; Pang et al., 2023; Perini et al., 2023), and unsupervised AD (Ruff et al., 2018; Ye et al., 2023). However, there hasn't been an effective AD algorithm tailored for the form where anomalous are both scarce and sparse.

## 3 DERIVATION OF UNBIASED AND BALANCED PU LEARNING

In classical supervised learning problems, we often deal with positive-negative (PN) learning. Assume there is an underlying distribution $p(x, y)$, where $x \in \mathbb{R}^d$ is the input, and $y \in \{-1, +1\}$ is the output. Data of size $n_+$ are sampled from $p(x|y = +1)$, and data of size $n_-$ are sampled from $p(x|y = -1)$. We let $g : \mathbb{R}^d \to \mathbb{R}^2$ be a decision function from a function space $\mathcal{G}$, where $g_{-1}$ and $g_{+1}$ are the probabilities of the sample being negaftive and positive, respectively. We also let $\mathcal{L} : \mathbb{R}^d \times \{-1, +1\} \to \mathbb{R}$ be a loss function. The goal of PN learning is to use P data and N data to learn a classifier, denoted as $f : \mathbb{R}^d \to \{-1, +1\}$ which is based on the decision function $g$.

However, in some scenarios, we may only have labels for one class, either only positive class labels known as PU learning or only negative class labels known as Negative-Unlabeled (NU) learning which is equivalent. In PU learning, data of size $n_+$ is sampled from $p(x|y = +1)$, and data of size $n_u$ is sampled from $p(x)$. The goal is also to learn a classifier $f : \mathbb{R}^d \to \{-1, +1\}$ which is the same as PN learning. We let $\pi = p(y = +1)$ represent the class prior of the positive data. The learning objective $R_{pn}(g) = E_{p(x,y)}[\mathcal{L}[(g(x), y]]$ of PN learning can be decomposed as

$$R_{pn}(g) = (1 - \pi)E_{p(x|y=-1)}[\mathcal{L}[g(x), -1]] + \pi E_{p(x|y=+1)}[\mathcal{L}[g(x), +1]]. \quad (1)$$

Du Plessis et al. (Du Plessis et al., 2015) derived an unbiased PU (uPU) learning objective. Kiryo et al. (Kiryo et al., 2017) propose non-negative PU (nnPU), avoiding the situation in uPU where the non-negative part of the loss is negative. In them, due to the absence of labeled negative samples for PU learning, the term $E_{p(x|y=-1)}[\mathcal{L}[g(x), -1]]$ cannot be directly estimated. However, since

$$E_{p(x)}[\mathcal{L}[g(x), -1]] = (1 - \pi)E_{p(x|y=-1)}[\mathcal{L}[g(x), -1]] + \pi E_{p(x|y=+1)}[\mathcal{L}[g(x), -1]], \quad (2)$$

where $\pi$ is the proportion of positive samples, $E_{p(x|y=-1)}[\mathcal{L}[g(x), -1]]$ can be indirectly estimated by leveraging the unlabeled data to estimate $E_{p(x)}[\mathcal{L}[g(x), -1)]]$ and using positive data to estimate $E_{p(x|y=+1)}[\mathcal{L}[g(x), -1]]$. So the empirical unbiased PU learning loss can be estimated using the current model's prediction $\hat{g}$ as

$$\hat{R}_{upu}(\hat{g}) = \frac{\pi}{n_p} \sum_{x_i \in P} \mathcal{L}[\hat{g}(x_i), +1] + \frac{1}{n_u} \sum_{x_i \in U} \mathcal{L}[\hat{g}(x_i), -1] - \frac{\pi}{n_p} \sum_{x_i \in P} \mathcal{L}[\hat{g}(x_i), -1]. \quad (3)$$

Su et al. (Su et al., 2021) argued that previous approaches struggle with balanced metrics. The objective of unbiased PU learning is to train a classifier that is unbiased when the class distribution of the test data matches that of the unlabeled data, rather than creating a balanced classifier. This can lead to poor performance for one of the classes, especially when $\pi$ is close to 0 or 1. In such cases, even if the model classifies all samples into a single class, achieving high accuracy, it does not meet the goals for real-world applications. Our objective is to learn a balanced classifier, despite the imbalance between positive and negative data in the unlabeled set. Building on the balanced PN learning objective, we aim to address this challenge:

$$R_{bpn}(g) = \frac{1}{2} E_{p(x|y=+1)}[\mathcal{L}[g(x), +1]] + \frac{1}{2} E_{p(x|y=-1)}[\mathcal{L}[g(x), -1]], \tag{4}$$

Through empirical estimation, we can get the balanced PU learning objective:

$$\hat{R}_{bpu}(\hat{g}) = \frac{1}{2n_p} \sum_{x_i \in P} \mathcal{L}[\hat{g}(x_i), +1] + \frac{1}{2n_u(1-\pi)} \sum_{x_i \in U} \mathcal{L}[\hat{g}(x_i), -1]$$
$$- \frac{\pi}{2n_p(1-\pi)} \sum_{x_i \in P} \mathcal{L}[\hat{g}(x_i), -1]. \tag{5}$$

Theoretically, the loss function most related to accuracy is the 0-1 loss. A surrogate loss function that is unbiased with respect to the 0-1 loss should satisfy the condition $\mathcal{L}[t, +1] + \mathcal{L}[t, -1] = 1$. We directly set $\mathcal{L}[\hat{g}(x_i), -1] = 1 - \mathcal{L}[\hat{g}(x_i), +1]$ in $\hat{R}_{bpu}(g)$, yielding a simplified expression:

$$\hat{R}_{bpu}(g) = \frac{1}{2n_p(1-\pi)} \sum_{x_i \in P} \mathcal{L}[\hat{g}(x_i), +1] + \frac{1}{2n_u(1-\pi)} \sum_{x_i \in U} \mathcal{L}[\hat{g}(x_i), -1] - \frac{\pi}{2(1-\pi)}. \tag{6}$$

This simple formula illustrates a straightforward yet counterintuitive principle. When the loss function satisfies $\mathcal{L}[t, +1] + \mathcal{L}[t, -1]$ as a constant, directly treating unlabeled samples as negative and training the model using the expected loss for each class supervisely results in a balanced learner. In other words, the model becomes unbiased with respect to the average accuracy metric. Building on this, we further derive the micro-level learning objective for PU learning.

## 4 BALANCED FINE-GRAINED PU LEARNING

### 4.1 MICRO-TO-MACRO OPTIMIZATION

In the problems we encounter, the key information that determines a sample as anomalous represents only a small portion of the anomalous samples. Specifically, all local components within the normal samples do not contain anomalous information, while in the anomalous samples, apart from a small amount of local anomalous information, the rest is normal information. We are given a macro dataset $D_{macro} = \{(X_1, Y_1), \ldots, (X_{|D_{macro}|}, Y_{|D_{macro}|})\}\}$. It can be represented separately as $P_{macro}$ and $N_{macro}$. For a macro sample of length $l$: $X_i = [x_{i1}, x_{i2}, \cdots, x_{il}]$, $x_{ij} \in \mathbb{R}^d$, $j \in [1, l]$ is the input, and $Y_i = F([y_{i1}, y_{i2}, \cdots, y_{il}])$, $y_{ij} \in \{-1, +1\}$ is the output where $F$ is the function that transforms micro labels into macro labels which constitutes a MIL problem (Carbonneau et al., 2018):

$$F([y_{i1}, \cdots, y_{il}]) = \begin{cases} +1, & \forall j \in \{1, \cdots, l\}, y_{ij} = +1 \\ -1, & \exists j \in \{1, \cdots, l\}, y_{ij} = -1 \end{cases} \tag{7}$$

We denote $G$ as the macro decision function which is transformed from the micro decision function $g$. Based on the relationship between the micro classifier $f$ and the macro classifier $F$, that is, within a set of macro data, if all micro-components belong to the normal class, the macro label is normal; if there exists at least one anomalous component, the macro label is anomalous. This can be reformulated as another description: if the micro component most inclined towards the anomalous class is anomalous, then the macro label is anomalous; otherwise, the macro label is normal. If such a micro component can be found through $g$ which is the idealized decision function, $G$ can be defined:

$$G([g(x_{i1}), g(x_{i2}), \cdots, g(x_{il})]) = \sum_{j=1}^{l} p(j = \arg\max_k g_{-1}(x_{ik})) g(x_{ij}) \tag{8}$$

Referring to eq. (8), we aim to find the idealized $p$ which makes $p(j = \arg\max_k g_{-1}(x_{ik})) = \mathbb{I}[j = \arg\max_k g_{-1}(x_{ik})]$. However, since we cannot obtain $g$, during the training process of neural networks, $\hat{g}$ from the neural network often deviates somewhat from the ideal decision function $g$. Consequently, we cannot directly obtain the value of the indicator function $\mathbb{I}[j = \arg\min_k g_{-1}(x_{ik})]$, but we can estimate the probability of the condition being satisfied. So, we use the probability function $\hat{p}$ instead of the function $p$.

$$\hat{p}(x_{ij}) = \frac{\exp(\hat{g}_{-1}(x_{ij}))}{\sum_{k=1}^{l} \exp(\hat{g}_{-1}(x_{ik}))} \tag{9}$$

Given our task, we are no longer pursuing the performance of the PU learning model at the micro level. Our ultimate goal is to ensure that the model trained at the micro level is beneficial for macro learning. We define the unbiased and balanced coarse-grained PN learning objective for the model as

$$\hat{R}_{cgpn}(\hat{g}) = \frac{1}{2 \cdot |P_{macro}| \cdot l} \sum_{X_i \in P_{macro}} \mathcal{L}[G([\hat{g}(x_{i1}), \cdots, \hat{g}(x_{il})]), +1]$$
$$+ \frac{1}{2 \cdot |N_{macro}| \cdot l} \sum_{X_i \in N_{macro}} \mathcal{L}[G([\hat{g}(x_{i1}), \cdots, \hat{g}(x_{il})]), -1] \tag{10}$$

Based on eqs. (6) and (8) to (10), through the transformation of the learning paradigm, we derive a loss function that enables unbiased and balanced optimization of macro-level performance directly at the micro level which is called balanced fine-grained PU learning loss:

$$\hat{R}_{cgpn}(\hat{g}) = \hat{R}_{bfgpu}(\hat{g}) = \frac{1}{2|P_{micro}|} \sum_{x_{ij} \in P_{micro}} \hat{p}(x_{ij}) \mathcal{L}[\hat{g}(x_{ij}), +1]$$
$$+ \frac{1}{2|U_{micro}|} \sum_{x_{ij} \in U_{micro}} \hat{p}(x_{ij}) \mathcal{L}[\hat{g}(x_{ij}), -1] \tag{11}$$

### 4.2 Pseudo Labels Based on Macro Information

After an epoch of PU learning, the model has acquired some discriminative ability for normal and anomalous samples. Further training with PN learning can be considered by obtaining pseudo-labels from unlabeled data. In the previous stage, we only utilized the micro information of the samples, neglecting the macro information. In the new stage, we can leverage this macro information. For an anomalous macro sample, since we know it contains at least one anomalous micro sample, we can directly label the most inclined-to-anomalous micro sample as an anomalous sample. Additionally, to ensure the balance of the learner, we simultaneously label the most inclined-to-normal micro sample as a normal sample. This way, we obtain an equal number of positive and negative pseudo-labeled samples for further model training. Expressed symbolically as:

$$N_{pse} = \{(x_{ij}, -1), j = \arg\max_k \hat{g}_{-1}(x_{ik}), X_i \in N_{macro}\} \tag{12}$$

$$P_{pse} = \{(x_{ij}, +1), j = \arg\min_k \hat{g}_{-1}(x_{ik}), X_i \in N_{macro}\} \tag{13}$$

Then we use $N_{pse}$ and $P_{pse}$ to further train the model. This ensures the improvement while maintaining the balance between positive and negative samples. The loss function is:

$$\hat{R}_{pse}(\hat{g}) = \sum_{(x_i, y_i) \in N_{pse} + P_{pse}} \mathcal{L}[\hat{g}(x_i), y_i] \tag{14}$$

### 4.3 Adjusted Decision Threshold (ADT)

After the training is completed, we deploy the model for testing. During the learning process, it is necessary to set an appropriate threshold. The final label is determined by comparing the decision function with the threshold. Typically, a default threshold of $0.5$ is assumed, but a more precise threshold can be determined by using the normal label distribution $\pi = 1 - \frac{1}{(\sigma_{micro}+1)\cdot(\sigma_{macro}+1)}$. Here, $\sigma_{micro} = (l-1)$ and $\sigma_{macro} = \frac{|P_{macro}|}{|N_{macro}|}$ denote the imbalance ratios of the MIL problem at

the micro and macro levels, respectively. Since we have access to the decision function values $\hat{g}(x)$, for all micro-unlabeled samples, it is sufficient to sort $\hat{g}(x)_{-1}$ in ascending order. After sorting, we can choose the position that corresponds to $\pi$ as the threshold $T$:

$$T = sort([\hat{g}_{-1}(x_i), x_i \in U_{micro}])[\lfloor |U_{micro}| \cdot \pi \rfloor] \tag{15}$$

This approach helps determine an accurate threshold because the decision function $g(x)$ reflects the conditional probability $p(y|x)$. $p(y|x)$ remains consistent between training and testing data. Therefore, the threshold obtained on the unlabeled data is highly applicable to the test data. The overall process of the algorithm can be seen in algorithm 1.

## 5 EXPERIMENTS

### 5.1 EXPERIMENTAL SETTINGS

For textual data, we primarily employ RoBERTa (Liu et al., 2019) as the backbone model (except for ablation studies concerning the backbone itself). For image data, we utilize ResNet-18 as the backbone to ensure a fair comparison across all algorithms. We uniformly adopt Adam as the optimizer (Kingma & Ba, 2015), with a learning rate set to $10^{-5}$, and CosineAnnealingLR as the learning rate scheduler. The number of epochs was set to 5, batch size to 16. We denote $\sigma_{micro}$ as the imbalance ratio at the micro level, which represents the positive-negative ratio of micro samples in negative macro samples. $\sigma_{macro}$ represents the imbalance ratio at the macro level, which is the ratio of positive macro samples to negative macro samples. We set $\lambda_{bfgpu} = 1/\pi$ and $\lambda_{pse} = 1$ in all experiments. For micro-level PU learning, $\pi = \frac{\sigma_{micro}}{\sigma_{micro}+1}$. The algorithms were implemented using the PyTorch framework (Paszke et al., 2019). We used average accuracy and F1 score, two commonly used evaluation metrics in imbalanced learning. Regarding the F1 score, when dealing with class imbalance problems, we typically focus on the performance of the minority (anomalous) class. We also compared the algorithm performance as $\sigma_{micro}$ varies, using the area under the curve $AUC_{AvgAcc}$ and $AUC_{F1}$. 4 A800 GPUs are used for all experiments.

### 5.2 COMPARED METHODS

Our comparative algorithms consist of: **Macro Supervised**: Conventional macro binary classification. **Macro DeepSAD**: Supervised macro AD using DeepSAD. (Ruff et al., 2019). **Macro Imbalanced Learning**: Under sampling and Over sampling. **MIL**: MIL algorithms used to address inexact supervision, using five multi-instance learning algorithms MIL-MaxPooling, MIL-TopkPooling (k was set to 3), MIL-Attention (Ilse et al., 2018), MIL-FGSA (Angelidis & Lapata, 2018), and MIL-PReNET (Pang et al., 2023). **Micro DeepSAD**: Supervised micro AD, using DeepSAD (Ruff et al., 2019) and treating unlabeled samples as negative class. **Micro DeepSVDD**: Unsupervised micro AD, using DeepSVDD (Ruff et al., 2018) which is an One-Class Classification (OCC) method using only positive samples. **Micro PU Learning**: Utilizing four types of loss functions uPU (Du Plessis et al., 2015), nnPU (Kiryo et al., 2017), balancedPU (Su et al., 2021), and robustPU (Zhu et al., 2023) for micro-level PU learning. **LLMs**: GPT-4-turbo (Achiam et al., 2023), Qwen3 (Yang et al., 2025), Deepseek v3 (Liu et al., 2024), Gemini2.5 (Comanici et al., 2025).

### 5.3 EXPERIMENTS WITH THE IDC DATASET

We evaluated various methods with a standard MIL dataset IDC, which is a medical image classification dataset primarily used for breast cancer detection research. This dataset contains patches of breast tissue images labeled according to the presence of invasive ductal carcinoma. In the dataset setup, each bag is a patient and the $\sigma_{macro}$ is around 3. Each instance is a 50×50 pixel image patch, and each bag contains 4 instances. The results are shown in table 1.

### 5.4 EXPERIMENTS WITH THE CSQI DATASET

We evaluated various methods with a real-world CSQI dataset depicted in fig. 1. The dataset consists of instances where service personnel were flagged for substandard performance during service. In CSQI, anomalous samples are extremely scarce, with only 24 dialogue sessions, while normal samples are abundant. we set $\sigma_{macro} = 100$ and randomly sampled 2,400 normal sessions for the experiments. The $\sigma_{micro}$ varied inconsistently, averaging around 25. The results are shown in table 2.

Table 1: The comparison on the IDC dataset.

| | Method | AvgAcc | F1 Score |
|---|---|---|---|
| Macro | Supervised | 58.70 | 57.13 |
| | DeepSAD | 44.30 | 43.20 |
| MIL | Maxpooling | 65.55 | 62.39 |
| | Topkpooling | 63.74 | 63.82 |
| | Attention | 56.03 | 49.10 |
| | FGSA | 60.22 | 59.86 |
| | PReNET | 49.30 | 38.46 |
| Micro | DeepSAD | 61.94 | 58.81 |
| | DeepSVDD | 52.15 | 49.10 |
| | uPU | 56.88 | 56.36 |
| | nnPU | 62.66 | 61.00 |
| | balancePU | 57.33 | 55.73 |
| | robustPU | 64.38 | 63.16 |
| | BFGPU | **73.87** | **73.21** |

Table 2: The comparison on the CSQI dataset.

| | Method | AvgAcc | F1 Score |
|---|---|---|---|
| Macro | Supervised | 52.38 | 8.33 |
| | DeepSAD | 47.61 | 6.67 |
| MIL | Maxpooling | 48.81 | 46.67 |
| | Topkpooling | 52.38 | 45.45 |
| | Attention | 54.76 | 53.27 |
| | FGSA | 53.57 | 54.14 |
| | PReNET | 59.52 | 59.05 |
| Micro | DeepSAD | 58.33 | 63.12 |
| | DeepSVDD | 50.00 | 0.00 |
| | uPU | 50.00 | 0.00 |
| | nnPU | 50.00 | 0.00 |
| | balancedPU | 50.00 | 0.00 |
| | robustPU | 50.00 | 0.00 |
| | BFGPU | **71.43** | **75.31** |

## 5.5 EXPERIMENTS WITH SYNTHETIC DATASETS

Since real datasets cannot control the imbalance ratios $\sigma_{macro}$ and $\sigma_{micro}$, we use sentiment analysis datasets to synthesize datasets with different imbalance ratios to explore the capability of various algorithms in handling dual imbalance problems. For example, we randomly combine 5 positive samples and 1 negative sample from the original dataset to synthesize a positive sample with $\sigma_{micro} = 5$. For short-text ones SST-2 (Socher et al., 2013) and Sentiment140 (Go et al., 2009), $\sigma_{micro}$ was set to $[2, 4, 6, 8, 10]$. For long-text datasets IMDB (Maas et al., 2011) and Amazon, $\sigma_{micro}$ was set to $[2, 3, 4, 5]$, the remaining experimental results can be found in. All experiments were repeated 3 times using random seeds $[0, 1, 2]$. The results can be found in tables 3 and 5 to 8. We considered a more extreme scenario where the normal-anomalous ratio at not only the micro

Table 3: The table presents the AvgAcc of various algorithms under varying values of $\sigma_{\mathrm{micro}} \in \{2, 4, 6, 8, 10\}$, along with the estimated AUC concerning changes in AvgAcc on the SST-2 Dataset.

| Methods | | AvgAcc | | | | | $AUC_{AvgAcc}$ |
|---|---|---|---|---|---|---|---|
| | | 2 | 4 | 6 | 8 | 10 | |
| Macro | Supervised | $83.15 \pm 0.98$ | $78.38 \pm 1.02$ | $76.57 \pm 3.42$ | $71.38 \pm 9.29$ | $73.26 \pm 2.51$ | 76.55 |
| | DeepSAD | $76.52 \pm 0.23$ | $59.41 \pm 6.75$ | $52.17 \pm 1.02$ | $49.69 \pm 3.47$ | $50.39 \pm 2.19$ | 57.64 |
| MIL | MaxPooling | $85.82 \pm 1.78$ | $66.83 \pm 6.44$ | $61.84 \pm 10.31$ | $52.83 \pm 4.90$ | $56.20 \pm 10.74$ | 64.70 |
| | TopkPooling | $-$ | $75.25 \pm 5.35$ | $75.85 \pm 2.74$ | $68.87 \pm 3.27$ | $64.73 \pm 12.81$ | 71.18 |
| | Attention | $80.76 \pm 1.57$ | $70.96 \pm 3.72$ | $57.49 \pm 12.35$ | $50.00 \pm 0.00$ | $51.16 \pm 2.01$ | 60.07 |
| | FGSA | $68.05 \pm 16.87$ | $50.00 \pm 0.00$ | $50.00 \pm 0.00$ | $50.00 \pm 0.00$ | $50.00 \pm 0.00$ | 53.61 |
| | PReNET | $78.82 \pm 6.43$ | $67.33 \pm 3.25$ | $66.49 \pm 3.32$ | $56.92 \pm 6.07$ | $60.47 \pm 3.49$ | 66.00 |
| Micro | DeepSAD | $61.33 \pm 5.75$ | $51.16 \pm 1.53$ | $48.79 \pm 1.49$ | $49.06 \pm 0.77$ | $49.22 \pm 0.55$ | 51.91 |
| | DeepSVDD | $48.62 \pm 1.93$ | $50.00 \pm 2.97$ | $49.52 \pm 0.42$ | $49.06 \pm 0.94$ | $50.00 \pm 1.16$ | 49.44 |
| | uPU | $80.02 \pm 1.92$ | $71.29 \pm 3.86$ | $69.81 \pm 2.80$ | $67.92 \pm 4.00$ | $53.88 \pm 2.74$ | 68.58 |
| | nnPU | $81.49 \pm 0.00$ | $50.17 \pm 0.00$ | $50.00 \pm 0.00$ | $50.00 \pm 0.00$ | $50.00 \pm 0.00$ | 56.33 |
| | balancedPU | $77.90 \pm 7.82$ | $75.74 \pm 8.49$ | $65.70 \pm 11.42$ | $54.72 \pm 3.85$ | $57.75 \pm 10.96$ | 66.36 |
| | robustPU | $86.28 \pm 1.39$ | $75.91 \pm 9.20$ | $74.15 \pm 6.16$ | $61.64 \pm 11.80$ | $55.42 \pm 9.40$ | 70.68 |
| | BFGPU | $\mathbf{88.40 \pm 0.68}$ | $\mathbf{82.51 \pm 0.62}$ | $\mathbf{82.13 \pm 0.90}$ | $\mathbf{79.56 \pm 1.60}$ | $\mathbf{82.56 \pm 1.64}$ | **83.03** |

but also the macro level is imbalanced. This lead to the anomalous information accounting for only $\frac{1}{(1+\sigma_{micro})\cdot(1+\sigma_{macro})}$. However, BFGPU still achieved outstanding performance. We set $\sigma_{macro}$ to 5 and 10 and conducted the experiments. We plotted the curve of average accuracy varying using synthetic SST-2 as shown in fig 3 and 4. The experimental results are shown in tables 9 and 10.

## 5.6 COMPARED WITH LLMS

We maximized the $\sigma_{micro}$ in the synthetic datasets, setting it to 5 for IMDB and Amazon, and to 10 for SST-2 and Sentiment140, to evaluate whether the model can accurately identify samples with slight anomalies. This further enabled a comparison of the "needle-in-a-haystack" capability between BFGPU and current mainstream LLMs. We used a consistent prompt: "You are a sentiment classification model; output 'positive' if there is no negative sentiment in the paragraph, and 'negative'

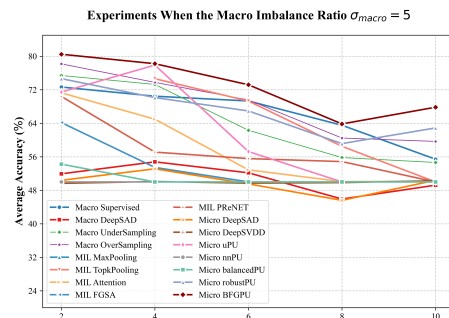
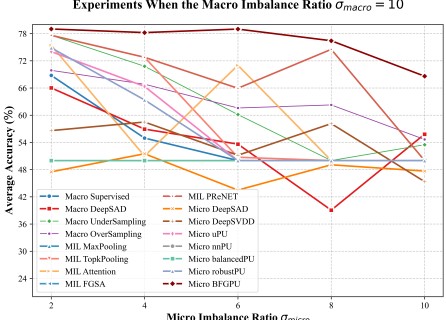

Figure 3: The figure presents the AvgAcc of various algorithms when $\sigma_{macro} = 5$.

Figure 4: The figure presents the AvgAcc of various algorithms when $\sigma_{macro} = 10$.

if negative sentiment exists. You must only output 'positive' or 'negative' without any additional content." The experimental results, shown in table 4, confirm that BFGPU is currently more suitable than LLMs themselves as a reward model in the RLHF process.

Table 4: The table presents the performance compared with LLMs with maximum $\sigma_{micro}$.

| Model | IMDB | | Amazon | | SST-2 | | Sentiment140 | |
|---|---|---|---|---|---|---|---|---|
| | AvgAcc | F1 Score | AvgAcc | F1 Score | AvgAcc | F1 Score | AvgAcc | F1 Score |
| GPT-4-turbo | 64.71 | 71.43 | 79.41 | 86.27 | 70.45 | 74.95 | 55.88 | 59.46 |
| Qwen3 | 67.65 | 66.67 | 82.35 | 86.36 | 67.86 | 64.52 | 64.71 | 62.50 |
| Deepseek v3 | 55.88 | 48.28 | 67.65 | 71.79 | 57.58 | 56.25 | 61.76 | 55.17 |
| Gemini2.5 | 64.71 | 62.50 | 79.41 | 85.11 | 63.09 | 74.42 | 58.82 | 61.11 |
| BFGPU | **83.64** | **84.61** | **85.57** | **86.50** | **82.56** | **84.41** | **66.45** | **66.40** |

### 5.7 ABLATION STUDY AND SENSITIVITY ANALYSIS

We conducted some ablation experiments. a) To validate the necessity of each component of BFGPU, we considered 3 settings: Not using the BFGPU learning loss function; Not using pseudo-labeling for further training; Not using adjusted thresholds. The results are shown in tables 11 and 12. b) To verify that our results are not limited by the choice of backbone model, we conducted further evaluations using DeBERTa table 13. The results are shown in table 13. c) To further demonstrate the superiority of the BFGPU loss function, we compared its performance against other PU learning methods when integrated with our proposed pseudo-labeling and ADT strategies. The results are shown in table 14.

BFGPU mainly has two hyperparameters, $\lambda_{bfgpu}$ and $\lambda_{pse}$. To validate the stability under different hyperparameter settings, we conducted a sensitivity analysis using the SST-2 dataset. The results of $\lambda_{bfgpu}$ and $\lambda_{pse}$ are shown in table 15 and table 16 respectively.

## 6 THEORETICAL ARGUMENTATION

We conducted analyses of the generalization error bounds for coarse-grained PN learning, MIL, and balanced fine-grained PU learning, identifying their respective applicability ranges. We also pointed out the necessity of adopting the micro PU learning paradigm, especially when the abnormal content is low. The theoretical analysis below is based on two assumptions (Niu et al., 2016):

**Assumption 1.** *There is a constant $C_{\mathcal{G}} > 0$ such that: $\mathcal{R}_{n,q}(\mathcal{G}) \leq C_{\mathcal{G}/\sqrt{n}}$ where $\mathcal{R}_{n,q}(\mathcal{G})$ is the Rademacher complexity of the function space $\mathcal{G}$ for $n$ samples from any marginal distribution $q(x)$.*

**Assumption 2.** *The loss function $L$ satisfies the symmetric condition and $\alpha_L$-Lipschitz continuity:*

$$L(t, +1) + L(t, -1) = 1, |L(t_1, y) - L(t_2, y)| \leq \alpha_L |t_1 - t_2| \tag{16}$$

At the macro level, the average generalization error bound for all classes under normal circumstances is relatively easy to infer. However, due to the issue of extremely low abnormal class information in abnormal class samples, as much as $\frac{l-1}{l}$ of the information in the classification task samples can be redundant or even considered as noise. Therefore, the generalization error bound of coarse-grained PN (CGPN) learning that we derive is:

**Theorem 1.** *For any $\delta > 0$, with probability at least $1 - \delta$:*

$$\hat{R}(g_{cgpn}) - R(g^*) \leq \frac{2 \cdot (\sigma_{macro} + 1) \cdot \sqrt{\sigma_{micro+1}} \cdot \alpha_L \cdot C_{\mathcal{G}}}{\sigma_{macro} \cdot \sqrt{|P_{micro}|}}$$

$$+ \frac{\sigma_{macro} + 1}{2 \cdot \sigma_{macro}} \cdot \sqrt{\frac{2 \cdot (\sigma_{micro} + 1) \cdot ln(4/\delta)}{|P_{micro}|}} + \frac{2 \cdot (\sigma_{macro} + 1) \cdot \sqrt{\sigma_{micro+1}} \cdot \alpha_L \cdot C_{\mathcal{G}}}{\sqrt{|U_{micro}|}}$$

$$+ \frac{\sigma_{macro} + 1}{2} \cdot \sqrt{\frac{2 \cdot (\sigma_{micro} + 1) \cdot ln(4/\delta)}{|U_{micro}|}} + Disc(\mathcal{G}, p_1(x, y), p_{1/l}(x, y)) \quad (17)$$

*where $g^* = \arg\min_{g \in \mathcal{G}} R(g)$ be the optimal decision function for $p_1(x, y)$ in $\mathcal{G}$, and*

$$Disc(\mathcal{G}, p_1(x, y), p_{1/l}) = \max_{g \in \mathcal{G}} |p_{x,y \sim p_1(x,y)}(g(x) \neq y) - p_{x,y \sim p_{1/l}(x,y)}(g(x) \neq y)| \quad (18)$$

*represents the discrepancy in distribution between the effective information proportion of $1/l$ and the effective information proportion of 1 for the function set $\mathcal{G}$.*

In classic MIL algorithms, all instances within an anomalous bag are treated as anomalous, which introduces significant bias. The extent of this bias is jointly determined by the imbalance levels at both the micro and macro levels, denoted as $\sigma_{micro}$ and $\sigma_{macro}$, respectively. Therefore, the generalization error bound of MIL that we derive is:

**Theorem 2.** *For any $\delta > 0$, with probability at least $1 - \delta$:*

$$\hat{R}(g_{mil}) - R(g^*) \leq \frac{2 \cdot (\sigma_{macro} + 1) \cdot \alpha_L \cdot C_{\mathcal{G}}}{\sigma_{macro} \cdot \sqrt{|P_{micro}|}} + \frac{2 \cdot (\sigma_{macro} + 1) \cdot \alpha_L \cdot C_{\mathcal{G}}}{\sqrt{|U_{micro}|}} + \frac{\sigma_{macro} + 1}{2 \cdot \sigma_{macro}}$$

$$\cdot \sqrt{\frac{2 \cdot ln(4/\delta)}{|P_{micro}|}} + \frac{\sigma_{macro} + 1}{2} \cdot \sqrt{\frac{2 \cdot ln(4/\delta)}{|U_{micro}|}} + p_{inc}(\mathcal{G}, g^*) + \frac{\sigma_{macro} + 1}{2} \cdot \frac{\sigma_{micro}}{\sigma_{micro} + 1} \quad (19)$$

*where $p_{inc}(\mathcal{G}, g^*) = \max_{g \in \mathcal{G}} p(\arg\max_j g_{-1}(x_j) \neq \arg\max_j g^*_{-1}(x_j))$ means the inconsistency between the true closest abnormal micro-sample in the macro sample and the predicted one.*

Based on the derivation of the PU loss in eq. (11), we found that directly optimizing unlabeled samples as abnormal samples essentially balances the loss of PN learning. Therefore, the generalization error bound of BFGPU that we derive is:

**Theorem 3.** *For any $\delta > 0$, with probability at least $1 - \delta$:*

$$\hat{R}(g_{bfgpu}) - R(g^*) \leq \frac{4 \cdot \alpha_L \cdot C_{\mathcal{G}}}{\sqrt{|P_{micro}|}} + \sqrt{\frac{2 \cdot ln(4/\delta)}{|P_{micro}|}} + \frac{4 \cdot \alpha_L \cdot C_{\mathcal{G}}}{\sqrt{|U_{micro}|}} + \sqrt{\frac{2 \cdot ln(4/\delta)}{|U_{micro}|}} + p_{inc}(\mathcal{G}, g^*)$$

$$(20)$$

The above theoretical results indicate that when the proportion of negative samples at the micro level is too low in the macro-level samples, the micro PU learning paradigm not only reduces the variance caused by model space complexity and confidence $\delta$ due to a large number of samples but also does not need to face the distribution discrepancy caused by a large amount of redundant or noisy information. BFGPU eliminates the bias in MIL, and unlike CGPN and MIL—which are heavily affected by $\sigma_{micro}$ and $\sigma_{macro}$—its error bound is not influenced by the class imbalance problem.

## 7 CONCLUSION

We addressed the challenge of scarce and sparse anomalous detection in real-world scenarios: macroscopically, there is high similarity between normal and anomalous samples, and microscopically, labels are lacking and imbalanced. Accordingly, we propose a dual-imbalanced MIL problem. We transformed the problem into an imbalanced PU learning task at the micro level, demonstrating its feasibility through theoretical analysis. We proposed a solution that directly optimizes macro-level performance at the micro level using PU learning objectives and combined pseudo-labeling with threshold adjustment techniques to create a new framework. Experiments validated that this framework effectively tackles the issue of sparse negative information while maintaining strong performance even in extreme cases of imbalance at both macro and micro levels.

ETHICS STATEMENT

We foresee no direct negative societal impacts resulting from this work, which is intended to advance research in computational music and creativity.

REPRODUCIBILITY STATEMENT

We are fully committed to the reproducibility of our research. The code for this work has been open-sourced at https://github.com/BFGPU/BFGPU, and the experimental setup section of the paper provides a comprehensive description of all models and parameters used. We will subsequently focus on providing a more user-friendly interface to facilitate their use of the proposed method as much as possible.

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

## A  LLMs Usage Statement

We used Large Language Models (LLMs) to assist with the writing of this paper. Their primary role was to improve grammar, phrasing, and clarity.

## B  Algorithm Framework

This section presents the specific algorithmic framework.

---

**Algorithm 1** Balanced Fine-Grained PU Learning

---

**Training Phase**

**Input:** macro positive dataset $P_{macro}$, macro negative dataset $N_{macro}$, the coefficient of $\hat{R}_{bfgpu}$ $\lambda_{bfgpu}$, the coefficient of $\hat{R}_{pse}$ $\lambda_{pse}$, learning rate $\eta$, the number of epochs $E$, class distribution prior $\pi$.

**Output:** micro classifier $g$, threshold $T$.

Split the macro-level data: $P_{micro} \leftarrow Split(P_{macro})$

Split the macro-level data: $U_{micro} \leftarrow Split(N_{macro})$

Initialize g with parameters $\theta$

**for** $e = 1$ **to** $E$ **do**

  **for** $P_{batch}, U_{batch}$ **in** $P_{micro}, U_{micro}$ **do**

    Get the probabilities $\hat{p}$ by eq. (9)

    Get the loss $\hat{R}_{bfgpu}$ by eq. (11)

    $\theta = \theta - \eta \nabla_\theta (\lambda_{bfgpu} \cdot \hat{R}_{bfgpu})$

  **end for**

  Get $P_{pse}, N_{pse}$ by eqs. (12) and (13)

  **for** $P_{batch}, N_{batch}$ **in** $P_{pse}, N_{pse}$ **do**

    Get the loss $\hat{R}_{pse}$ by eq. (14)

    $\theta = \theta - \eta \nabla_\theta (\lambda_{pse} \cdot \hat{R}_{pse})$

  **end for**

**end for**

Get the threshold $T$ by eq. (15)

**Testing Phase**

**Input:** macro test data $X_T$, micro classifier $g$, and threshold $T$.

**Output:** macro prediction $Y_T$.

**for** $X_i$ **in** $X_T$ **do**

  Initial $Y_{Ti} \leftarrow +1$

  **for** $x_{ij}$ **in** $X_i$ **do**

    **if** $g_{-1}(x_{ij}) > T$ **then**

      Update $Y_{Ti} \leftarrow -1$

    **end if**

  **end for**

**end for**

---

## C  Proof of the Theories

### C.1  Proof of Theorem 4.1

Consider directly learning from macro-level data at the macro level, where the proportion of effective information is $1/l$ in the macro, and then testing on data with the same proportion. It can be shown

that there exists $\delta > 0$, with at least a probability of $1 - \delta$:

$$\hat{R}(g_{cgpn}) - R(g^*_{1/l})$$

$$\leq \frac{1}{2} \cdot \left( \frac{4 \cdot \alpha_L \cdot C_{\mathcal{G}}}{\sqrt{|P_{micro}|/(\sigma_{micro} + 1)}} + \sqrt{\frac{2 \cdot ln(4/\delta)}{|P_{micro}|/(\sigma_{micro} + 1)}} \right) / \frac{\sigma_{macro}}{\sigma_{macro} + 1}$$

$$+ \frac{1}{2} \cdot \left( \frac{4 \cdot \alpha_L \cdot C_{\mathcal{G}}}{\sqrt{|U_{micro}|/(\sigma_{micro} + 1)}} + \sqrt{\frac{2 \cdot ln(4/\delta)}{|U_{micro}|/(\sigma_{micro} + 1)}} \right) / \frac{1}{\sigma_{macro} + 1} \qquad (21)$$

Then, considering the data distribution caused by redundant information differs from the data distribution when there is no redundant information, it is not difficult to conclude that:

$$R(g^*_{1/l}) - R_(g^*) \leq Disc(\mathcal{G}, p_1(x, y), p_{1/l}(x, y)) \qquad (22)$$

By combining the two equations, we can measure the gap between the macro PN error and the error of the optimal classifier at the macro level when there is no redundant information. Ultimately proved for any $\delta > 0$, with probability at least $1 - \delta$:

$$\hat{R}(g_{cgpn}) - R(g^*)$$

$$= (\hat{R}(g_{cgpn}) - R(g^*_{1/l})) + (R(g^*_{1/l}) - R_(g^*))$$

$$\leq \frac{2 \cdot (\sigma_{macro} + 1) \cdot \sqrt{\sigma_{micro+1}} \cdot \alpha_L \cdot C_{\mathcal{G}}}{\sigma_{macro} \cdot \sqrt{|P_{micro}|}}$$

$$+ \frac{\sigma_{macro} + 1}{2 \cdot \sigma_{macro}} \cdot \sqrt{\frac{2 \cdot (\sigma_{micro} + 1) \cdot ln(4/\delta)}{|P_{micro}|}} + \frac{2 \cdot (\sigma_{macro} + 1) \cdot \sqrt{\sigma_{micro+1}} \cdot \alpha_L \cdot C_{\mathcal{G}}}{\sqrt{|U_{micro}|}}$$

$$+ \frac{\sigma_{macro} + 1}{2} \cdot \sqrt{\frac{2 \cdot (\sigma_{micro} + 1) \cdot ln(4/\delta)}{|U_{micro}|}} + Disc(\mathcal{G}, p_1(x, y), p_{1/l}(x, y)) \qquad (23)$$

### C.2 PROOF OF THEOREM 4.2

Consider assigning the macro-level labels to all micro-level samples. For normal samples, all training labels are correct; whereas for anomalous samples, the upper bound of labeling error is $\frac{\sigma_{micro}}{\sigma_{micro}+1}$. When the class imbalance ratio at the macro level is $\sigma_{macro}$, it is straightforward to derive the balanced error upper bound for training with the MIL method. It can be shown that there exists $\delta > 0$, with at least a probability of $1 - \delta$:

$$\hat{R}(g_{mil}) - R(g^*_{micro})$$

$$\leq \frac{1}{2} \cdot \left( \frac{4 \cdot \alpha_L \cdot C_{\mathcal{G}}}{\sqrt{|P_{micro}|}} + \sqrt{\frac{2 \cdot ln(4/\delta)}{|P_{micro}|}} \right) / \frac{\sigma_{macro}}{\sigma_{macro} + 1}$$

$$+ \frac{1}{2} \cdot \left( \frac{4 \cdot L_l \cdot C_{\mathcal{G}}}{\sqrt{|U_{micro}|}} + \sqrt{\frac{2 \cdot ln(4/\delta)}{|U_{micro}|}} + \frac{\sigma_{micro}}{\sigma_{micro} + 1} \right) / \frac{1}{\sigma_{macro} + 1} \qquad (24)$$

Considering the inconsistency between micro-level optimization goals and macro-level optimization goals, it is not difficult to conclude that:

$$R(g^*_{micro}) - R(g^*) \leq p_{inc}(\mathcal{G}, g^*) \qquad (25)$$

By combining the two equations, we can measure the gap between the MIL error and the error of the optimal classifier at the macro level. Ultimately proved for any $\delta > 0$, with probability at least $1 - \delta$: For any $\delta > 0$, with probability at least $1 - \delta$:

$$\hat{R}(g_{mil}) - R(g^*)$$

$$= (\hat{R}(g_{mil}) - R(g_{micro}^*)) + (R(g_{micro}^*) - R(g^*))$$

$$\leq \frac{2 \cdot (\sigma_{macro} + 1) \cdot \alpha_L \cdot C_{\mathcal{G}}}{\sigma_{macro} \cdot \sqrt{|P_{micro}|}} + \frac{2 \cdot (\sigma_{macro} + 1) \cdot \alpha_L \cdot C_{\mathcal{G}}}{\sqrt{|U_{micro}|}} + \frac{\sigma_{macro} + 1}{2 \cdot \sigma_{macro}} \cdot \sqrt{\frac{2 \cdot ln(4/\delta)}{|P_{micro}|}}$$

$$+ \frac{\sigma_{macro} + 1}{2} \cdot \sqrt{\frac{2 \cdot ln(4/\delta)}{|U_{micro}|}} + p_{inc}(\mathcal{G}, g^*) + \frac{\sigma_{macro} + 1}{2} \cdot \frac{\sigma_{micro}}{\sigma_{micro} + 1} \tag{26}$$

### C.3 Proof of Theorem 4.3

By directly optimizing the balanced PU loss at the micro level, we can achieve unbiased optimization of macro-level performance. Taking into account the error introduced by converting the macro-level problem to the micro level, we ultimately obtain an error upper bound for BFGPU that is independent of both imbalance ratios $\sigma_{macro}$ and $\sigma_{micro}$. It can be shown that there exists $\delta > 0$, with at least a probability of $1 - \delta$:

$$\hat{R}(g_{bfgpu}) - R(g_{micro}^*)$$

$$\leq \frac{4 \cdot L_l \cdot C_{\mathcal{G}}}{\sqrt{|P_{micro}|}} + \frac{4 \cdot L_l \cdot C_{\mathcal{G}}}{\sqrt{|U_{micro}|}}$$

$$+ \sqrt{\frac{2 \cdot ln(4/\delta)}{|P_{micro}|}} + \sqrt{\frac{2 \cdot ln(4/\delta)}{|U_{micro}|}} \tag{27}$$

Considering the inconsistency between micro-level optimization goals and macro-level optimization goals, it is not difficult to conclude that:

$$R(g_{micro}^*) - R(g^*) \leq p_{inc}(\mathcal{G}, g^*) \tag{28}$$

By combining the two equations, we can measure the gap between the micro PU error and the error of the optimal classifier in the marco level when there is no redundant information. Ultimately proved for any $\delta > 0$, with probability at least $1 - \delta$: For any $\delta > 0$, with probability at least $1 - \delta$:

$$\hat{R}(g_{bfgpu}) - R(g^*)$$

$$= (\hat{R}(g_{bfgpu}) - R(g_{micro}^*)) + (R(g_{micro}^*) - R(g^*))$$

$$\leq \frac{4 \cdot L_l \cdot C_{\mathcal{G}}}{\sqrt{|P_{micro}|}} + \frac{4 \cdot L_l \cdot C_{\mathcal{G}}}{\sqrt{|U_{micro}|}}$$

$$+ \sqrt{\frac{2 \cdot ln(4/\delta)}{|P_{micro}|}} + \sqrt{\frac{2 \cdot ln(4/\delta)}{|U_{micro}|}} + p_{inc}(\mathcal{G}, g^*) \tag{29}$$

## D Additional Experimental Results

It is evident that our proposed BFGPU achieves optimal performance in most settings. In only a few cases does the nnPU or robustPU loss function achieve optimal performance, but this also depends on our proposed ADT technique.

### D.1 Experiments with Synthetic Datasets

In addition to the SST-2 dataset, we conducted experiments on Sentiment140 which is a large short-text dataset with a focus on high imbalance ratios, setting $\sigma_{micro}$ to [2, 4, 6, 8, 10].

Due to space limitations in the main text, the additional experiments on the SST-2 and Sentiment140 datasets are provided in tables 5 and 6 in this appendix. We present a comparative experiment on

Table 5: The table presents the F1 scores of various algorithms under varying values of $\sigma_{\mathrm{micro}} \in \{2, 4, 6, 8, 10\}$, along with the estimated AUC with respect to changes in F1 on the SST-2 Dataset.

| Methods | | F1 Score | | | | | $AUC_{F1}$ |
| | | 2 | 4 | 6 | 8 | 10 | |
|---|---|---|---|---|---|---|---|
| Macro | Supervised | $83.63 \pm 0.21$ | $79.35 \pm 1.04$ | $78.15 \pm 2.81$ | $75.77 \pm 5.79$ | $75.50 \pm 3.23$ | 78.48 |
| | DeepSAD | $75.24 \pm 1.86$ | $47.14 \pm 13.51$ | $28.13 \pm 9.66$ | $13.04 \pm 9.22$ | $7.00 \pm 7.06$ | 34.11 |
| MIL | MaxPooling | $85.28 \pm 7.74$ | $52.34 \pm 4.00$ | $37.42 \pm 3.35$ | $12.38 \pm 4.36$ | $19.49 \pm 4.03$ | 41.38 |
| | TopkPooling | $-$ | $69.95 \pm 6.85$ | $70.39 \pm 4.89$ | $61.75 \pm 4.28$ | $44.44 \pm 38.49$ | 61.63 |
| | Attention | $80.54 \pm 0.51$ | $62.39 \pm 7.22$ | $45.00 \pm 36.50$ | $0.00 \pm 0.00$ | $48.79 \pm 30.96$ | 47.34 |
| | FGSA | $49.38 \pm 43.89$ | $44.44 \pm 38.49$ | $66.67 \pm 0.00$ | $44.44 \pm 38.49$ | $44.44 \pm 38.49$ | 49.87 |
| | PReNET | $73.67 \pm 10.48$ | $53.84 \pm 7.58$ | $48.68 \pm 6.08$ | $48.77 \pm 16.00$ | $36.10 \pm 12.41$ | 52.21 |
| Micro | DeepSAD | $51.95 \pm 12.30$ | $14.95 \pm 2.18$ | $9.01 \pm 7.18$ | $9.92 \pm 2.40$ | $14.18 \pm 4.29$ | 20.00 |
| | DeepSVDD | $24.98 \pm 20.38$ | $12.08 \pm 5.99$ | $26.98 \pm 19.65$ | $10.96 \pm 1.51$ | $13.30 \pm 4.16$ | 17.66 |
| | uPU | $76.82 \pm 3.40$ | $64.43 \pm 6.62$ | $62.13 \pm 6.88$ | $60.79 \pm 10.15$ | $67.56 \pm 1.27$ | 66.35 |
| | nnPU | $84.00 \pm 1.25$ | $66.74 \pm 0.10$ | $66.67 \pm 0.00$ | $66.67 \pm 0.00$ | $66.67 \pm 0.00$ | 70.15 |
| | balancedPU | $80.75 \pm 4.89$ | $78.73 \pm 6.27$ | $52.51 \pm 37.17$ | $61.83 \pm 7.17$ | $70.48 \pm 5.39$ | 68.86 |
| | robustPU | $86.31 \pm 0.72$ | $76.10 \pm 6.57$ | $74.79 \pm 3.74$ | $63.29 \pm 10.26$ | $62.90 \pm 6.51$ | 72.68 |
| | BFGPU | $\mathbf{88.73 \pm 0.61}$ | $\mathbf{84.13 \pm 0.30}$ | $\mathbf{83.96 \pm 1.10}$ | $\mathbf{81.95 \pm 1.87}$ | $\mathbf{84.41 \pm 1.64}$ | $\mathbf{84.64}$ |

Table 6: The table presents AvgAcc and F1 scores of various algorithms under varying values of $\sigma_{\mathrm{micro}} \in \{2, 4, 6, 8, 10\}$, along with the estimated AUC with respect to changes in AvgAcc and F1 on the Sentiment140 Dataset.

| Method | | AvgAcc | | | | | $AUC_{AvgAcc}$ |
| | | 2 | 4 | 6 | 8 | 10 | |
|---|---|---|---|---|---|---|---|
| Macro | Supervised | $74.37 \pm 0.40$ | $69.80 \pm 0.37$ | $66.81 \pm 0.48$ | $65.09 \pm 0.19$ | $62.20 \pm 0.48$ | 67.65 |
| | DeepSAD | $57.03 \pm 4.17$ | $53.41 \pm 0.94$ | $52.56 \pm 1.24$ | $51.37 \pm 0.63$ | $51.21 \pm 0.99$ | 53.12 |
| MIL | MaxPooling | $60.79 \pm 2.63$ | $62.51 \pm 9.54$ | $56.78 \pm 5.84$ | $49.95 \pm 0.09$ | $52.03 \pm 3.41$ | 56.41 |
| | TopkPooling | $-$ | $65.08 \pm 1.90$ | $62.14 \pm 2.28$ | $56.31 \pm 5.96$ | $52.95 \pm 5.07$ | 59.12 |
| | Attention | $71.27 \pm 0.36$ | $64.97 \pm 3.24$ | $52.88 \pm 4.98$ | $50.06 \pm 0.10$ | $50.16 \pm 0.28$ | 57.87 |
| | FGSA | $64.17 \pm 3.13$ | $53.46 \pm 4.03$ | $50.00 \pm 0.00$ | $50.00 \pm 0.00$ | $50.00 \pm 0.00$ | 53.53 |
| | PReNET | $70.41 \pm 3.74$ | $57.11 \pm 7.11$ | $55.56 \pm 4.83$ | $54.85 \pm 2.93$ | $50.00 \pm 0.00$ | 57.59 |
| Micro | DeepSAD | $51.07 \pm 2.14$ | $51.80 \pm 2.05$ | $65.23 \pm 2.04$ | $63.99 \pm 8.28$ | $58.81 \pm 8.64$ | 58.18 |
| | DeepSVDD | $49.65 \pm 0.55$ | $50.09 \pm 0.18$ | $49.70 \pm 0.58$ | $49.83 \pm 0.72$ | $50.27 \pm 0.29$ | 49.91 |
| | uPU | $68.81 \pm 1.02$ | $62.73 \pm 2.48$ | $61.23 \pm 3.25$ | $57.54 \pm 5.37$ | $54.31 \pm 3.33$ | 60.92 |
| | nnPU | $69.66 \pm 1.01$ | $50.00 \pm 0.00$ | $50.00 \pm 0.00$ | $50.00 \pm 0.00$ | $50.00 \pm 0.00$ | 53.93 |
| | balancedPU | $72.55 \pm 2.23$ | $70.26 \pm 0.16$ | $66.16 \pm 1.53$ | $50.00 \pm 0.00$ | $50.68 \pm 0.54$ | 61.93 |
| | robustPU | $74.65 \pm 0.70$ | $70.13 \pm 1.63$ | $66.94 \pm 0.40$ | $59.14 \pm 8.58$ | $62.88 \pm 0.27$ | 66.74 |
| | BFGPU | $\mathbf{75.86 \pm 1.01}$ | $\mathbf{70.68 \pm 0.97}$ | $\mathbf{67.95 \pm 1.53}$ | $\mathbf{66.34 \pm 1.25}$ | $\mathbf{64.45 \pm 0.35}$ | $\mathbf{69.06}$ |

| Method | | F1 Score | | | | | $AUC_{F1}$ |
| | | 2 | 4 | 6 | 8 | 10 | |
|---|---|---|---|---|---|---|---|
| Macro | Supervised | $73.99 \pm 0.72$ | $68.75 \pm 0.44$ | $64.94 \pm 1.66$ | $63.86 \pm 0.52$ | $59.50 \pm 1.48$ | 66.21 |
| | DeepSAD | $57.24 \pm 4.26$ | $53.61 \pm 1.16$ | $52.03 \pm 1.33$ | $51.55 \pm 0.71$ | $51.83 \pm 0.77$ | 53.25 |
| MIL | MaxPooling | $37.85 \pm 7.23$ | $43.23 \pm 30.95$ | $27.71 \pm 23.78$ | $21.92 \pm 37.96$ | $9.96 \pm 16.53$ | 28.13 |
| | TopkPooling | $-$ | $52.83 \pm 4.60$ | $47.32 \pm 4.56$ | $26.12 \pm 23.20$ | $34.85 \pm 33.37$ | 40.28 |
| | Attention | $67.64 \pm 0.52$ | $56.56 \pm 5.58$ | $37.40 \pm 34.07$ | $22.85 \pm 37.96$ | $23.34 \pm 37.56$ | 41.56 |
| | FGSA | $52.19 \pm 8.97$ | $16.28 \pm 17.59$ | $44.44 \pm 38.49$ | $44.44 \pm 38.49$ | $44.44 \pm 38.49$ | 40.36 |
| | PreNET | $63.36 \pm 8.65$ | $26.89 \pm 24.61$ | $44.62 \pm 19.17$ | $20.65 \pm 12.24$ | $66.67 \pm 0.00$ | 44.44 |
| Micro | DeepSAD | $54.65 \pm 1.80$ | $55.89 \pm 1.76$ | $67.50 \pm 1.80$ | $66.55 \pm 7.42$ | $61.86 \pm 7.61$ | 61.29 |
| | DeepSVDD | $53.33 \pm 0.67$ | $54.27 \pm 0.20$ | $53.37 \pm 0.71$ | $53.61 \pm 1.05$ | $54.11 \pm 0.49$ | 53.74 |
| | uPU | $60.88 \pm 2.42$ | $57.32 \pm 10.29$ | $48.65 \pm 13.84$ | $60.66 \pm 4.54$ | $66.66 \pm 0.21$ | 58.83 |
| | nnPU | $75.50 \pm 0.56$ | $66.67 \pm 0.00$ | $66.67 \pm 0.00$ | $66.67 \pm 0.00$ | $66.67 \pm 0.00$ | 68.44 |
| | balancedPU | $72.00 \pm 1.80$ | $69.98 \pm 0.95$ | $70.80 \pm 0.84$ | $44.44 \pm 31.43$ | $\mathbf{66.83 \pm 0.24}$ | 64.81 |
| | robustPU | $73.43 \pm 1.05$ | $67.66 \pm 1.83$ | $61.80 \pm 2.36$ | $64.00 \pm 2.35$ | $55.89 \pm 0.48$ | 64.55 |
| | BFGPU | $\mathbf{77.21 \pm 0.73}$ | $\mathbf{73.27 \pm 0.95}$ | $\mathbf{69.80 \pm 3.18}$ | $\mathbf{67.21 \pm 2.94}$ | $66.40 \pm 0.68$ | $\mathbf{70.78}$ |

two long text datasets: IMDB and Amazon. The parameter $\sigma_{micro}$ is set to [2, 3, 4, 5]. Due to space limitations in the main text, the additional experiments on the datasets Amazon are shown in tables 7 and 8 in this appendix. It is noticeable that when the imbalance ratio is high, the performance

Table 7: The table presents the AvgAcc and F1 scores of various algorithms under varying values of $\sigma_{\text{micro}} \in \{2, 3, 4, 5\}$, along with the estimated AUC on the IMDB Dataset.

| Method | | AvgAcc | | | | $AUC_{AvgAcc}$ |
|---|---|---|---|---|---|---|
| | | 2 | 3 | 4 | 5 | |
| Macro | Supervised | $76.75 \pm 1.45$ | $70.62 \pm 0.83$ | $62.61 \pm 4.70$ | $58.33 \pm 4.81$ | 67.08 |
| | DeepSAD | $52.17 \pm 0.56$ | $52.57 \pm 1.20$ | $55.01 \pm 1.05$ | $50.76 \pm 0.77$ | 52.63 |
| MIL | MaxPooling | $85.76 \pm 2.55$ | $74.30 \pm 18.50$ | $60.31 \pm 15.71$ | $55.56 \pm 9.63$ | 68.98 |
| | TopkPooling | $-$ | $82.23 \pm 1.22$ | $83.27 \pm 0.61$ | $82.41 \pm 2.17$ | 82.63 |
| | Attention | $82.58 \pm 7.37$ | $85.98 \pm 1.32$ | $80.66 \pm 3.81$ | $78.62 \pm 1.54$ | 81.96 |
| | FGSA | $82.67 \pm 1.60$ | $63.60 \pm 13.43$ | $60.49 \pm 14.59$ | $58.64 \pm 14.97$ | 66.35 |
| | PReNET | $84.53 \pm 1.50$ | $80.54 \pm 1.50$ | $71.78 \pm 4.97$ | $73.56 \pm 2.49$ | 77.60 |
| Micro | DeepSAD | $62.86 \pm 4.33$ | $60.00 \pm 1.43$ | $57.80 \pm 1.46$ | $56.84 \pm 0.12$ | 59.38 |
| | DeepSVDD | $52.70 \pm 0.61$ | $51.73 \pm 1.13$ | $51.96 \pm 0.09$ | $51.76 \pm 2.58$ | 52.04 |
| | uPU | $82.93 \pm 1.14$ | $76.30 \pm 1.51$ | $78.57 \pm 5.37$ | $72.29 \pm 3.30$ | 77.52 |
| | nnPU | $84.53 \pm 1.49$ | $58.39 \pm 5.95$ | $50.00 \pm 0.00$ | $50.00 \pm 0.00$ | 60.73 |
| | balancedPU | $85.78 \pm 1.39$ | $77.39 \pm 3.90$ | $84.85 \pm 1.14$ | $78.77 \pm 4.17$ | 81.70 |
| | robustPU | $87.82 \pm 0.37$ | $86.63 \pm 1.11$ | $85.49 \pm 0.45$ | $72.58 \pm 19.57$ | 83.13 |
| | BFGPU | $\mathbf{88.13 \pm 1.07}$ | $\mathbf{87.93 \pm 0.58}$ | $\mathbf{86.02 \pm 0.69}$ | $\mathbf{83.64 \pm 0.78}$ | **86.43** |

| Method | | F1 Score | | | | $AUC_{F1}$ |
|---|---|---|---|---|---|---|
| | | 2 | 3 | 4 | 5 | |
| Macro | Supervised | $77.73 \pm 2.22$ | $73.77 \pm 0.92$ | $63.98 \pm 10.18$ | $50.24 \pm 22.06$ | 66.43 |
| | DeepSAD | $52.22 \pm 0.59$ | $52.91 \pm 1.37$ | $54.84 \pm 1.33$ | $50.82 \pm 1.21$ | 52.70 |
| MIL | MaxPooling | $84.89 \pm 3.51$ | $59.75 \pm 41.13$ | $29.08 \pm 41.11$ | $20.76 \pm 35.96$ | 48.62 |
| | TopkPooling | $-$ | $79.15 \pm 1.95$ | $81.37 \pm 1.30$ | $80.26 \pm 2.95$ | 80.26 |
| | Attention | $82.27 \pm 7.82$ | $85.67 \pm 1.41$ | $78.78 \pm 4.93$ | $75.43 \pm 2.50$ | 80.54 |
| | FGSA | $81.50 \pm 3.14$ | $39.66 \pm 36.71$ | $52.30 \pm 30.53$ | $46.00 \pm 39.91$ | 54.87 |
| | PReNET | $82.83 \pm 2.11$ | $77.07 \pm 2.40$ | $61.39 \pm 9.70$ | $65.83 \pm 4.80$ | 71.78 |
| Micro | DeepSAD | $65.45 \pm 3.58$ | $62.67 \pm 1.56$ | $60.51 \pm 1.16$ | $59.78 \pm 0.85$ | 62.10 |
| | DeepSVDD | $55.45 \pm 1.36$ | $55.99 \pm 1.44$ | $55.99 \pm 0.60$ | $55.14 \pm 2.11$ | 55.64 |
| | uPU | $80.95 \pm 2.18$ | $71.34 \pm 2.77$ | $74.05 \pm 8.51$ | $63.94 \pm 6.10$ | 72.57 |
| | nnPU | $86.15 \pm 1.03$ | $70.68 \pm 2.85$ | $66.67 \pm 0.00$ | $66.67 \pm 0.00$ | 72.54 |
| | balancedPU | $85.94 \pm 1.78$ | $79.12 \pm 3.57$ | $85.69 \pm 0.93$ | $80.93 \pm 3.70$ | 82.92 |
| | robustPU | $87.52 \pm 0.53$ | $86.24 \pm 1.29$ | $85.06 \pm 0.79$ | $77.80 \pm 9.68$ | 84.15 |
| | BFGPU | $\mathbf{88.12 \pm 1.24}$ | $\mathbf{88.36 \pm 0.58}$ | $\mathbf{86.57 \pm 0.69}$ | $\mathbf{84.61 \pm 0.43}$ | **86.92** |

improvement brought by BFGPU is even greater compared to previous experiments. With larger values of $\sigma_{micro}$, we observed that many algorithms lack stability and may even fail. When algorithms fail, the evaluation metric F1 Score becomes highly unstable. This is because F1 Score does not treat positive and negative classes equally; more fundamentally, precision and recall are both centered around the positive class. Thus, the results in F1 Score can vary significantly depending on whether the classification leans towards the positive or negative class to the same extent. In contrast, the metric of average accuracy possesses greater stability and fairness.

## D.2 EXPERIMENTS WITH IMBALANCE AT BOTH MACRO AND MICRO LEVELS

On the SST-2 dataset, we set $\sigma_{macro}$ to 5 and 10 and explored how algorithm performance varied with $\sigma_{micro}$ set to [2, 4, 6, 8, 10].

Due to space limitations in the main text, the experimental statistical data on the SST-2 dataset are provided in tables 9 and 10 in this appendix. In settings where both imbalances coexist, with the total proportion of negative information in the text dataset reaching its extreme low, we found that most comparative methods failed. However, BFGPU still maintained stable and excellent performance in this extreme scenario, demonstrating its strong versatility.

Table 8: The table presents AvgAcc and F1 scores of various algorithms under varying values of $\sigma_{\text{micro}} \in \{2, 3, 4, 5\}$, along with the estimated AUC with respect to changes in AvgAcc and F1 on the Amazon Dataset.

| | Method | AvgAcc | | | | $AUC_{AvgAcc}$ |
|---|---|---|---|---|---|---|
| | | 2 | 3 | 4 | 5 | |
| Macro | Supervised | $89.11 \pm 0.19$ | $87.82 \pm 0.23$ | $84.74 \pm 0.95$ | $81.97 \pm 0.60$ | 85.91 |
| | DeepSAD | $85.55 \pm 0.43$ | $71.88 \pm 6.80$ | $63.31 \pm 1.65$ | $60.83 \pm 3.43$ | 70.39 |
| MIL | MaxPooling | $87.09 \pm 4.28$ | $85.25 \pm 3.01$ | $82.38 \pm 1.18$ | $78.96 \pm 4.81$ | 83.42 |
| | TopkPooling | $-$ | $86.15 \pm 0.11$ | $84.12 \pm 0.89$ | $82.59 \pm 1.74$ | 84.27 |
| | Attention | $87.96 \pm 0.93$ | $86.37 \pm 1.23$ | $85.60 \pm 0.18$ | $69.13 \pm 15.83$ | 82.27 |
| | FGSA | $59.46 \pm 16.39$ | $64.73 \pm 14.45$ | $52.50 \pm 4.33$ | $50.00 \pm 0.00$ | 56.67 |
| | PReNET | $87.15 \pm 0.68$ | $84.00 \pm 2.50$ | $77.88 \pm 2.86$ | $79.40 \pm 1.56$ | 82.11 |
| Micro | DeepSAD | $62.86 \pm 9.94$ | $52.14 \pm 0.99$ | $51.20 \pm 2.28$ | $50.84 \pm 0.17$ | 54.26 |
| | DeepSVDD | $50.2 \pm 1.36$ | $51.19 \pm 0.74$ | $50.45 \pm 1.06$ | $50.44 \pm 0.59$ | 50.60 |
| | uPU | $82.48 \pm 0.38$ | $76.92 \pm 2.66$ | $74.15 \pm 2.31$ | $74.69 \pm 0.18$ | 77.06 |
| | nnPU | $85.11 \pm 1.40$ | $74.02 \pm 5.18$ | $50.00 \pm 0.00$ | $50.00 \pm 0.00$ | 64.78 |
| | balancedPU | $80.59 \pm 7.70$ | $81.69 \pm 5.52$ | $80.84 \pm 6.93$ | $82.91 \pm 8.14$ | 81.51 |
| | robustPU | $89.28 \pm 0.52$ | $87.07 \pm 0.63$ | $86.34 \pm 0.34$ | $82.47 \pm 1.08$ | 86.29 |
| | BFGPU | $\mathbf{89.87 \pm 0.13}$ | $\mathbf{88.15 \pm 0.97}$ | $\mathbf{86.71 \pm 0.18}$ | $\mathbf{85.57 \pm 0.67}$ | **87.58** |

| | Method | F1 Score | | | | $AUC_{F1}$ |
|---|---|---|---|---|---|---|
| | | 2 | 3 | 4 | 5 | |
| Macro | Supervised | $89.10 \pm 0.20$ | $87.86 \pm 0.26$ | $84.73 \pm 0.95$ | $82.15 \pm 0.51$ | 85.96 |
| | DeepSAD | $88.22 \pm 0.25$ | $85.87 \pm 0.15$ | $83.02 \pm 0.60$ | $78.63 \pm 0.93$ | 83.94 |
| MIL | MaxPooling | $85.81 \pm 6.10$ | $82.82 \pm 4.57$ | $79.60 \pm 1.87$ | $74.56 \pm 8.00$ | 80.70 |
| | TopkPooling | $-$ | $84.80 \pm 0.17$ | $82.30 \pm 1.31$ | $80.26 \pm 2.73$ | 82.45 |
| | Attention | $88.24 \pm 0.91$ | $86.73 \pm 1.33$ | $86.16 \pm 0.15$ | $65.79 \pm 19.75$ | 81.73 |
| | FGSA | $47.01 \pm 20.89$ | $63.63 \pm 14.12$ | $10.08 \pm 17.43$ | $44.44 \pm 38.49$ | 41.29 |
| | PReNET | $86.10 \pm 0.88$ | $81.97 \pm 3.58$ | $72.85 \pm 4.66$ | $75.66 \pm 2.73$ | 79.14 |
| Micro | DeepSAD | $85.47 \pm 0.54$ | $72.34 \pm 6.42$ | $63.52 \pm 1.48$ | $59.47 \pm 3.18$ | 70.20 |
| | DeepSVDD | $47.01 \pm 40.89$ | $63.63 \pm 14.12$ | $10.08 \pm 17.43$ | $44.44 \pm 38.49$ | 41.29 |
| | uPU | $79.82 \pm 0.62$ | $71.43 \pm 4.48$ | $67.49 \pm 4.27$ | $68.79 \pm 3.65$ | 71.88 |
| | nnPU | $86.63 \pm 1.09$ | $79.34 \pm 3.11$ | $66.67 \pm 0.00$ | $66.67 \pm 0.00$ | 74.83 |
| | balancedPU | $78.50 \pm 11.50$ | $81.24 \pm 6.09$ | $81.38 \pm 6.31$ | $83.76 \pm 0.71$ | 81.22 |
| | robustPU | $89.06 \pm 0.71$ | $86.60 \pm 0.83$ | $85.85 \pm 0.70$ | $82.18 \pm 1.25$ | 85.92 |
| | BFGPU | $\mathbf{90.03 \pm 0.08}$ | $\mathbf{88.51 \pm 1.05}$ | $\mathbf{87.44 \pm 0.17}$ | $\mathbf{86.50 \pm 0.40}$ | **88.12** |

Table 9: Experimental Results with Imbalance at Both Macro and Micro Levels when $\sigma_{macro=5}$.

| | Method | AvgAcc | | | | | $AUC_{AvgAcc}$ |
|---|---|---|---|---|---|---|---|
| | | 2 | 4 | 6 | 8 | 10 | |
| Macro | Supervised | 72.65 | 70.46 | 69.32 | 63.52 | 55.42 | 66.27 |
| | DeepSAD | 51.93 | 54.79 | 52.17 | 45.91 | 49.22 | 50.80 |
| | UnderSampling | 75.41 | 73.27 | 62.32 | 55.81 | 54.64 | 64.29 |
| | OverSampling | 78.18 | 73.76 | 69.57 | 60.47 | 59.67 | 68.33 |
| MIL | MaxPooling | 50.00 | 50.00 | 50.00 | 50.00 | 57.87 | 51.57 |
| | TopkPooling | – | 74.64 | 69.31 | 58.49 | 50.00 | 63.11 |
| | Attention | 77.90 | 71.29 | 51.21 | 56.29 | 58.14 | 62.96 |
| | FGSA | 50.00 | 50.00 | 50.00 | 50.00 | 50.00 | 50.00 |
| | PReNET | 50.00 | 50.00 | 50.00 | 50.00 | 50.00 | 50.00 |
| Micro | DeepSAD | 50.37 | 53.13 | 49.52 | 45.60 | 50.39 | 49.80 |
| | DeepSVDD | 51.38 | 54.29 | 50.00 | 48.74 | 53.10 | 51.50 |
| | uPU | 71.45 | 77.89 | 57.25 | 50.00 | 50.00 | 61.32 |
| | nnPU | 50.00 | 50.00 | 50.00 | 50.00 | 50.00 | 50.00 |
| | balancedPU | 54.24 | 50.00 | 50.00 | 50.00 | 50.00 | 50.85 |
| | robustPU | 75.60 | 76.57 | 58.21 | 50.00 | 50.00 | 62.08 |
| | BFGPU | **80.48** | **78.22** | **73.19** | **63.84** | **67.83** | **72.71** |

| | Method | F1 Score | | | | | $AUC_{F1}$ |
|---|---|---|---|---|---|---|---|
| | | 2 | 4 | 6 | 8 | 10 | |
| Macro | Supervised | 78.16 | 76.95 | 75.92 | 72.56 | 68.00 | 74.32 |
| | DeepSAD | 56.88 | 59.33 | 44.46 | 36.40 | 23.49 | 44.11 |
| | UnderSampling | 69.83 | 75.00 | 71.28 | 69.84 | 56.86 | 68.56 |
| | OverSampling | 70.56 | 65.79 | 62.37 | 60.47 | 62.90 | 64.42 |
| MIL | MaxPooling | 66.67 | 66.67 | 66.67 | 66.67 | 66.67 | 66.67 |
| | TopkPooling | – | 75.00 | 69.38 | 68.00 | 66.67 | 69.76 |
| | Attention | 81.46 | 77.75 | 67.21 | **69.70** | 70.73 | 73.37 |
| | FGSA | 66.67 | 66.67 | 66.67 | 66.67 | 66.67 | 66.67 |
| | PReNET | 66.67 | 66.67 | 66.67 | 66.67 | 66.67 | 66.67 |
| Micro | DeepSAD | 47.48 | 51.85 | 44.26 | 38.50 | 47.01 | 45.82 |
| | DeepSVDD | 48.99 | 51.81 | 44.85 | 48.33 | 50.36 | 48.87 |
| | uPU | 77.09 | 78.81 | 68.25 | 66.67 | 66.67 | 71.50 |
| | nnPU | 66.67 | 66.67 | 66.67 | 66.67 | 66.67 | 66.67 |
| | balancedPU | 13.85 | 0.00 | 0.00 | 0.00 | 0.00 | 2.77 |
| | robustPU | 80.07 | 80.74 | 71.03 | 66.67 | 66.67 | 73.04 |
| | BFGPU | **83.48** | **81.59** | **78.76** | 64.92 | **74.24** | **76.60** |

Table 10: Experimental Results with Imbalance at Both Macro and Micro Levels when $\sigma_{macro=10}$.

| | Method | AvgAcc | | | | | $AUC_{AvgAcc}$ |
|---|---|---|---|---|---|---|---|
| | | 2 | 4 | 6 | 8 | 10 | |
| Macro | Supervised | 68.78 | 54.95 | 50.00 | 50.00 | 50.00 | 54.75 |
| | DeepSAD | 66.02 | 56.93 | 53.62 | 39.06 | 55.81 | 54.29 |
| | UnderSampling | 77.62 | 70.79 | 60.14 | 50.00 | 53.48 | 62.41 |
| | OverSampling | 69.89 | 66.83 | 61.59 | 62.26 | 54.65 | 63.04 |
| MIL | MaxPooling | 50.00 | 50.00 | 50.00 | 50.00 | 50.00 | 50.00 |
| | TopkPooling | – | 72.77 | 50.72 | 50.00 | 50.00 | 55.87 |
| | Attention | 75.41 | 50.99 | 71.01 | 50.00 | 50.00 | 59.48 |
| | FGSA | 50.00 | 50.00 | 50.00 | 50.00 | 50.00 | 50.00 |
| | PReNET | 77.62 | 72.77 | 65.94 | 74.52 | 50.00 | 68.17 |
| Micro | DeepSAD | 47.51 | 51.49 | 43.48 | 49.06 | 47.67 | 47.84 |
| | DeepSVDD | 56.60 | 58.49 | 51.16 | 58.14 | 45.35 | 53.95 |
| | uPU | 74.03 | 66.34 | 50.00 | 50.00 | 50.00 | 58.07 |
| | nnPU | 50.00 | 50.00 | 50.00 | 50.00 | 50.00 | 50.00 |
| | balancedPU | 50.00 | 50.00 | 50.00 | 50.00 | 50.00 | 50.00 |
| | robustPU | 74.86 | 63.37 | 50.00 | 50.00 | 50.00 | 57.65 |
| | **BFGPU** | **79.01** | **78.22** | **78.99** | **76.42** | **68.60** | **76.25** |

| | Method | F1 Score | | | | | $AUC_{F1}$ |
|---|---|---|---|---|---|---|---|
| | | 2 | 4 | 6 | 8 | 10 | |
| Macro | Supervised | 75.80 | 68.51 | 66.67 | 66.67 | 66.67 | 68.86 |
| | DeepSAD | 74.43 | 65.32 | 53.62 | 56.88 | 32.14 | 56.48 |
| | UnderSampling | 77.56 | 63.80 | 49.54 | 66.67 | 23.08 | 56.13 |
| | OverSampling | 78.87 | 70.73 | 74.36 | 76.19 | 66.67 | 73.36 |
| MIL | MaxPooling | 0.00 | 0.00 | 0.00 | 0.00 | 0.00 | 0.00 |
| | TopkPooling | – | 78.43 | 66.99 | 66.67 | 66.67 | 69.69 |
| | Attention | 79.91 | 67.11 | 77.27 | 66.67 | 66.67 | 71.53 |
| | FGSA | 66.67 | 66.67 | 66.67 | 66.67 | 66.67 | 66.67 |
| | PReNET | 81.29 | 79.39 | 78.26 | 74.59 | 66.67 | 76.04 |
| Micro | DeepSAD | 53.66 | 59.17 | 50.63 | 51.79 | 60.18 | 55.09 |
| | DeepSVDD | 64.26 | 44.75 | 59.63 | 63.83 | 67.19 | 59.93 |
| | uPU | 83.56 | 71.67 | 66.67 | 66.67 | 66.67 | 71.05 |
| | nnPU | 66.67 | 66.67 | 66.67 | 66.67 | 66.67 | 66.67 |
| | balancedPU | 0.00 | 0.00 | 0.00 | 0.00 | 0.00 | 0.00 |
| | robustPU | 79.55 | 73.19 | 66.67 | 66.67 | 66.67 | 70.55 |
| | **BFGPU** | **87.27** | **81.48** | **78.52** | **76.19** | **67.19** | **78.13** |

## D.3 ABLATION STUDY

To explore the necessity of each component of BFGPU, we conducted ablation experiments, comparing the algorithm's performance without the new loss function, without pseudo-labeling, and without the threshold adjustment technique. We performed these experiments on the IMDB and SST-2 datasets, setting $\sigma_{micro}$ to [2, 4, 6, 8, 10]. Due to space limitations in the main text, the additional experiments on the SST-2 dataset are provided in table 12 in this appendix. The experimental results indicate that each component of BFGPU is crucial, collectively ensuring the algorithm's stability and excellent performance.

## D.4 EXPERIMENTS WITH DIFFERENT BASE MODEL

To verify that our results are not limited by the choice of backbone model, we conducted further evaluations using DeBERTa. The results are shown in table 13.

## D.5 EXPERIMENTS WITH DIFFERENT PU LOSS WITH PSEUDO LABELING AND ADT

o further demonstrate the superiority of the BFGPU loss function, we compared its performance against other PU learning methods when integrated with our proposed pseudo-labeling and ADT strategies. The results are shown in table 14.

Table 11: The table presents AvgAcc and F1 results of the ablation study under varying values of $\sigma_{\mathrm{micro}} \in \{2, 3, 4, 5\}$ on the IMDB dataset.

| PU Loss | Pseudo Labels | Threshold | AvgAcc | | | |
|---|---|---|---|---|---|---|
| | | | 2 | 3 | 4 | 5 |
| × | ✓ | ✓ | $84.98 \pm 3.14$ | $84.27 \pm 2.12$ | $85.55 \pm 0.31$ | $62.19 \pm 10.22$ |
| ✓ | × | ✓ | $88.13 \pm 0.76$ | $86.04 \pm 0.66$ | $85.40 \pm 0.18$ | $83.02 \pm 1.35$ |
| ✓ | ✓ | × | $\mathbf{88.25 \pm 0.90}$ | $86.97 \pm 0.89$ | $85.40 \pm 0.18$ | $81.18 \pm 2.13$ |
| ✓ | ✓ | ✓ | $88.13 \pm 1.07$ | $\mathbf{87.93 \pm 0.58}$ | $\mathbf{86.02 \pm 0.69}$ | $\mathbf{83.64 \pm 0.78}$ |

| PU Loss | Pseudo Labels | Threshold | F1 Score | | | |
|---|---|---|---|---|---|---|
| | | | 2 | 3 | 4 | 5 |
| × | ✓ | ✓ | $82.34 \pm 2.83$ | $84.06 \pm 2.37$ | $85.67 \pm 0.43$ | $58.35 \pm 13.32$ |
| ✓ | × | ✓ | $\mathbf{88.63 \pm 0.55}$ | $86.95 \pm 0.42$ | $85.01 \pm 0.88$ | $84.36 \pm 0.74$ |
| ✓ | ✓ | × | $88.44 \pm 0.97$ | $86.50 \pm 1.12$ | $85.01 \pm 0.88$ | $78.92 \pm 3.39$ |
| ✓ | ✓ | ✓ | $88.12 \pm 1.24$ | $\mathbf{88.36 \pm 0.58}$ | $\mathbf{86.57 \pm 0.69}$ | $\mathbf{84.61 \pm 0.43}$ |

Table 12: The table presents AvgAcc and F1 results of the ablation study under varying values of $\sigma_{\mathrm{micro}} \in \{2, 4, 6, 8, 10\}$ on the SST-2 dataset.

| PU Loss | Pseudo Labels | Threshold | AvgAcc | | | | |
|---|---|---|---|---|---|---|---|
| | | | 2 | 4 | 6 | 8 | 10 |
| × | ✓ | ✓ | $85.27 \pm 1.50$ | $83.33 \pm 0.84$ | $79.95 \pm 2.46$ | $77.36 \pm 2.67$ | $79.07 \pm 3.80$ |
| ✓ | × | ✓ | $87.75 \pm 2.08$ | $\mathbf{86.14 \pm 1.07}$ | $71.50 \pm 14.69$ | $66.35 \pm 12.76$ | $80.62 \pm 3.33$ |
| ✓ | ✓ | × | $86.83 \pm 0.13$ | $72.11 \pm 12.07$ | $74.15 \pm 1.49$ | $58.81 \pm 5.78$ | $60.85 \pm 7.91$ |
| ✓ | ✓ | ✓ | $\mathbf{88.40 \pm 0.68}$ | $82.51 \pm 0.62$ | $\mathbf{82.13 \pm 0.90}$ | $\mathbf{79.56 \pm 1.60}$ | $\mathbf{82.56 \pm 1.64}$ |

| PU Loss | Pseudo Labels | Threshold | F1 Score | | | | |
|---|---|---|---|---|---|---|---|
| | | | 2 | 4 | 6 | 8 | 10 |
| × | ✓ | ✓ | $86.65 \pm 0.92$ | $84.95 \pm 0.64$ | $82.74 \pm 1.33$ | $80.87 \pm 2.12$ | $81.32 \pm 2.96$ |
| ✓ | × | ✓ | $88.41 \pm 1.80$ | $\mathbf{87.31 \pm 0.95}$ | $61.39 \pm 31.32$ | $51.26 \pm 36.66$ | $82.65 \pm 2.92$ |
| ✓ | ✓ | × | $86.88 \pm 0.54$ | $71.80 \pm 10.90$ | $68.84 \pm 2.64$ | $40.53 \pm 8.76$ | $57.33 \pm 12.78$ |
| ✓ | ✓ | ✓ | $\mathbf{88.73 \pm 0.61}$ | $84.13 \pm 0.30$ | $\mathbf{83.96 \pm 1.10}$ | $\mathbf{81.95 \pm 1.87}$ | $\mathbf{84.41 \pm 1.64}$ |

Table 13: The table presents AvgAcc and F1 scores of various algorithms with the Deberta model under varying values of $\sigma_{\text{micro}} \in \{2, 4, 6, 8, 10\}$, along with the estimated AUC with respect to changes in AvgAcc and F1 on the SST-2 Dataset.

| | Method | AvgAcc | | | | | $AUC_{AvgAcc}$ |
|---|---|---|---|---|---|---|---|
| | | 2 | 4 | 6 | 8 | 10 | |
| Macro | Supervised | 81.22 | 77.72 | 78.26 | 77.36 | 77.91 | 78.49 |
| | DeepSAD | 49.17 | 59.41 | 42.75 | 50.94 | 47.67 | 49.99 |
| MIL | MaxPooling | 75.97 | 70.30 | 76.81 | 65.09 | 50.00 | 67.63 |
| | TopkPooling | – | 74.26 | 77.54 | 78.30 | 73.26 | 75.84 |
| | Attention | 72.65 | 65.35 | 52.90 | 50.83 | 50.94 | 58.53 |
| | FGSA | 73.48 | 64.36 | 54.35 | 50.94 | 50.00 | 58.63 |
| | PReNET | 71.82 | 61.39 | 65.22 | 55.66 | 61.63 | 63.14 |
| Micro | DeepSAD | 48.34 | 49.51 | 51.45 | 51.89 | 52.33 | 50.70 |
| | DeepSVDD | 48.07 | 40.10 | 52.90 | 51.89 | 51.16 | 48.82 |
| | uPU | 72.65 | 57.43 | 72.46 | 59.43 | 67.44 | 65.88 |
| | nnPU | 72.65 | 50.50 | 50.00 | 50.00 | 50.00 | 54.63 |
| | balancedPU | 82.32 | 80.20 | 78.26 | 76.42 | 72.93 | 78.03 |
| | robustPU | 83.98 | 80.20 | 76.47 | 83.33 | 80.19 | 80.50 |
| | BFGPU | **88.40** | **81.68** | **78.99** | **83.96** | **81.40** | **82.89** |

| | Method | F1 Score | | | | | $AUC_{F1}$ |
|---|---|---|---|---|---|---|---|
| | | 2 | 4 | 6 | 8 | 10 | |
| Macro | Supervised | 80.79 | 77.61 | 79.73 | 76.92 | 79.12 | 78.83 |
| | DeepSAD | 55.34 | 69.40 | 56.83 | 60.61 | 62.81 | 61.00 |
| MIL | MaxPooling | 65.37 | 63.42 | 59.76 | 50.04 | 53.70 | 58.46 |
| | TopkPooling | – | 63.23 | 64.87 | 53.44 | 55.46 | 59.00 |
| | FGSA | 65.71 | 55.56 | 22.22 | 3.70 | 0.00 | 29.44 |
| | Attention | 63.47 | 47.76 | 35.29 | 23.08 | 10.71 | 36.06 |
| | PReNET | 74.75 | 40.00 | 47.83 | 25.40 | 37.74 | 45.14 |
| Micro | DeepSAD | 63.55 | 66.23 | 57.32 | 61.65 | 67.72 | 63.29 |
| | DeepSVDD | 64.26 | 44.75 | 59.63 | 63.83 | 67.19 | 59.93 |
| | uPU | 64.52 | 33.85 | 66.67 | 41.10 | 62.16 | 53.66 |
| | nnPU | 78.15 | 66.89 | 66.67 | 66.67 | 66.67 | 69.01 |
| | balancedPU | 84.08 | 82.91 | 81.71 | 79.67 | 81.19 | 81.91 |
| | robustPU | 82.53 | 77.78 | 80.77 | 82.44 | 75.32 | 79.77 |
| | BFGPU | **88.65** | **84.12** | **81.76** | **85.96** | **83.33** | **84.76** |

Table 14: The table presents the performance of various PU learning methods when integrated with our proposed pseudo-labeling and ADT strategies.

| Method | AvgAcc | | | | | $AUC_{AvgAcc}$ |
|---|---|---|---|---|---|---|
| | 2 | 4 | 6 | 8 | 10 | |
| uPU | $82.69 \pm 2.02$ | $76.07 \pm 1.82$ | $71.26 \pm 3.36$ | $64.47 \pm 0.89$ | $61.24 \pm 4.78$ | 71.15 |
| nnPU | $87.02 \pm 0.60$ | $\mathbf{83.83 \pm 1.42}$ | $79.23 \pm 1.49$ | $75.16 \pm 2.48$ | $74.03 \pm 3.95$ | 79.85 |
| balancedPU | $79.83 \pm 6.66$ | $77.23 \pm 5.73$ | $68.84 \pm 11.79$ | $66.04 \pm 5.82$ | $72.09 \pm 1.64$ | 72.81 |
| robustPU | $86.00 \pm 1.84$ | $74.09 \pm 20.01$ | $78.02 \pm 2.33$ | $72.96 \pm 8.02$ | $61.63 \pm 14.52$ | 74.54 |
| BFGPU | $\mathbf{88.40 \pm 0.68}$ | $82.51 \pm 0.62$ | $\mathbf{82.13 \pm 0.90}$ | $\mathbf{79.56 \pm 1.60}$ | $\mathbf{82.56 \pm 1.64}$ | **83.03** |

| Method | F1 Scroe | | | | | $AUC_{F1}$ |
|---|---|---|---|---|---|---|
| | 2 | 4 | 6 | 8 | 10 | |
| uPU | $84.09 \pm 1.08$ | $77.99 \pm 1.55$ | $75.59 \pm 2.66$ | $69.91 \pm 1.50$ | $65.42 \pm 9.19$ | 74.60 |
| nnPU | $87.75 \pm 0.60$ | $\mathbf{85.13 \pm 0.84}$ | $81.56 \pm 1.00$ | $78.85 \pm 2.79$ | $78.39 \pm 2.57$ | 82.34 |
| balancedPU | $80.94 \pm 6.40$ | $79.54 \pm 4.81$ | $57.02 \pm 32.67$ | $65.03 \pm 13.16$ | $76.11 \pm 1.94$ | 71.73 |
| robustPU | $86.91 \pm 1.48$ | $59.86 \pm 45.37$ | $80.23 \pm 2.51$ | $74.05 \pm 8.97$ | $48.39 \pm 42.41$ | 69.89 |
| BFGPU | $\mathbf{88.73 \pm 0.61}$ | $84.13 \pm 0.30$ | $\mathbf{83.96 \pm 1.10}$ | $\mathbf{81.95 \pm 1.87}$ | $\mathbf{84.41 \pm 1.64}$ | **84.64** |

## D.6 SENSITIVE ANALYSIS

We have supplemented the sensitivity analysis of hyperparameters $\lambda_{bfgpu}$ and $\lambda_{pse}$. We completed experiments on the IMDB dataset, setting $\sigma_{micro} = 5$. We set the variation range of the two parameters to [1, 2, 3, 4, 5], and kept one constant while varying the other. Due to space limitations in the main text, the sensitive analysis of $\lambda_{pse}$ on the SST-2 dataset are provided in table 16 in this appendix. Experiments have demonstrated that the performance of the algorithm remains

Table 15: This table presents the AvgAcc of the sensitive analysis of $\lambda_{bfgpu}$ on the SST-2 dataset.

| $\lambda_{bfgpu}$ | 1 | 2 | 3 | 4 | 5 |
|---|---|---|---|---|---|
| $AvgAcc$ | 83.41 | 84.72 | 84.99 | 84.55 | 82.79 |
| $F1Score$ | 84.56 | 85.68 | 85.83 | 85.56 | 84.45 |

Table 16: This table presents the AvgAcc of the sensitive analysis of $\lambda_{pse}$ on the SST-2 dataset.

| $\lambda_{pse}$ | 1 | 2 | 3 | 4 | 5 |
|---|---|---|---|---|---|
| $AvgAcc$ | 83.41 | 84.29 | 84.68 | 84.64 | 84.94 |
| $F1Score$ | 84.56 | 85.23 | 85.49 | 85.51 | 85.67 |

relatively stable under different hyperparameter settings, so there is no need to worry too much about hyperparameter tuning.

