# OpenReview forum: "Detecting Scarce and Sparse Anomalous: Solving Dual Imbalance in Multi-Instance Learning"
_ICLR.cc/2026/Conference — ICLR 2026 Conference Withdrawn Submission_

### Official Review · Reviewer_9uQv · 2025-10-29

**Soundness:** 3
**Presentation:** 3
**Contribution:** 3
**Rating:** 4
**Confidence:** 3

**Summary:**

This paper addresses the detection of scarce, sparse anomalies that cause dual imbalance (macro- and micro-level) in Multi-Instance Learning. It reformulates MIL as a fine-grained Positive-Unlabeled (PU) learning task, derives a balanced PU loss via rigorous theoretical analysis, and proposes the BFGPU framework that incorporates macro-level information to assign high-confidence pseudo-labels and dynamically adjust thresholds. Extensive experiments on synthetic datasets, real IDC (medical) and CSQI (customer service) datasets, plus comparisons with supervised/AD/MIL/PU methods and LLMs, validate BFGPU’s effectiveness with theoretical error bound analysis.

**Strengths:**

1. It clearly formalizes the "dual-imbalanced MIL problem" (macro-level scarce anomalous samples, micro-level sparse anomalous information), addressing a gap in traditional MIL that only focuses on single-level imbalance.

2. This paper reformulates MIL as a fine-grained PU learning task, integrating macro information into micro learning to realize micro-to-macro performance optimization.

3. The authors conduct systematic experiments (synthetic + real datasets like IDC/CSQI) and compare with multi-type baselines (supervised, MIL, PU, LLMs), plus ablation/parameter sensitivity analyses, effectively verifying method validity.

**Weaknesses:**

1. The paper reformulates MIL as fine-grained PU learning but does not distinguish it from existing MIL-PU hybrid methods explicitly. For example, PUMA, proposed by Perini et al. (2023), already explored "learning from positive and unlabeled multi-instance bags in anomaly detection," and PUMA also assigns instance-level labels. The paper does not clarify how its MIL-to-PU reformulation differs in essence from prior work, thereby weakening the uniqueness of its theoretical motivation. Additionally, PUMA is not compared in the experiment.

2. Though Line 83 claims to "dynamically adjust the confidence threshold to maintain balanced prediction", the critical threshold-related parameter π is fixed—it depends on pre-set σ_micro = l -1 and σ_macro (π=σ_micro/(σ_micro+1) and π=1/((σ_micro+1)(σ_macro+1)), contradicting the stated dynamic adjustment.

3. Additionally, while the paper aims for equal positive/negative pseudo-labels by selecting "most anomaly-inclined" and "most normal-inclined" instances from anomalous bags, it provides no solution for scenarios where \(\hat{g}\)’s probabilities are all extremely large/small (e.g., negligible differences between "most inclined" instances or uniform class inclination).

4. The paper provides no details on whether comparative methods underwent consistent hyperparameter optimization, which is critical for fair performance evaluation. It specifies tuning for BFGPU (e.g., λ_bfgpu, λ_pse) but does not confirm whether the baselines were optimized within the same search space.

5. The paper’s two θ updates per epoch (raw micro data + pseudo-labels) inherently increase computational overhead, yet it provides no efficiency comparisons with baselines. For real-time applications (e.g., online CSQI), metrics such as training time per epoch or inference latency are critical.

**Questions:**

Same to Weakness.

---

> ### Author Response · Authors · 2025-12-01
>
> Dear Reviewer, we hope the following response can address your concerns.
>
> ---
>
> **W1:**
> There are two reasons why we cited PUMA but did not include it in our comparisons:
>
> 1. PUMA is designed for a **PU MIL** setting, where bags at the coarse level contain only *positive* and *unlabeled* labels. Our work, instead, uses a fine-grained PU algorithm to solve a coarse-grained $PN MIL$ problem (i.e., standard PN MIL or imbalanced PN MIL). In other words, PUMA treats PU as a *bag-level* MIL problem setting, while in our work PU learning is an *instance-level* tool for solving MIL. These two setups are fundamentally different.
> 2. PUMA only works on tabular data and relies on the distribution of featureized variables with explicitly defined attributes. Our experiments are primarily conducted on text and image data, where PUMA cannot be applied due to the complexity of these modalities.
>
> ---
>
> **W2:**
>
> 1. Our “dynamic adjustment” refers to adjusting the threshold based on the changing output distribution of the neural network predictor $\hat{g}$.
> 2. The value of $\pi$ is determined by the data in practical applications—for instance, the number of sentences in a paragraph or the number of patches an image is split into. In our synthetic dataset experiments, $\pi$ was predefined only because we needed to control different imbalance ratios to verify the generality of the algorithm under various scenarios.
>
> ---
>
> **W3:**
> Our balancing is performed **within anomalous bags**, and we focus specifically on scenarios where anomalies are **sparse**. This means anomalous bags naturally contain a large number of normal instances and a small number of anomalous instances, with substantial distinguishability among them. For example, in diseased organs, both healthy and cancerous regions coexist, and different regions exhibit varying degrees of abnormality.
>
> ---
>
> **W4:**
> Our hyperparameter stability analysis is **not** a hyperparameter search for optimal performance. Instead, it is meant to verify that the performance is not overly sensitive to hyperparameters. The loss weight used in comparisons is consistently set rather than optimized in the stability experiments. For other algorithms in comparison, we follow the **optimal hyperparameters reported in their original papers**.
>
> ---
>
> **W5:**
> Because we only select a **very small amount** of additional data for balancing, the time overhead remains roughly linear relative to the baseline. Theoretically, pseudo-labeling introduces an additional computational cost of about $2 / (\sigma_{\text{micro}} + 1)$, which is quite small in practice due to the sparsity of the target tasks. For example, when $\sigma_{\text{micro}} = 10$, one epoch takes approximately **1153s** for the baseline and **1368s** for our method. In online CSQI-style tasks, such additional computation to handle **both scarcity and sparsity** is fully justified.
>
> Thank you again for taking the time to review our work.

---

### Official Review · Reviewer_w1pY · 2025-10-30

**Soundness:** 1
**Presentation:** 1
**Contribution:** 1
**Rating:** 2
**Confidence:** 5

**Summary:**

The paper bridging ideas from MIL and PU learning. It proposes BFGPU to address the imbalance problem at bag-level (macro) and instance-level (micro). The method combines a balanced PU loss, pseudo-label refinement, and an adaptive decision threshold to handle extreme class imbalance and uncertain pseudo-positives. The authors provide theoretical generalization bounds. Experiments show that BFGPU outperforms baseline PU and MIL methods in F1 and average accuracy, particularly under imbalanced regimes.

**Strengths:**

- Motivation is strong.
- Baselines are comprehensive.
- Evaluations on vision and language tasks are comprehensive and have high practical values.

**Weaknesses:**

- My main concern is about the novelty. MIL is already studied for anomaly detection in connection to learning with unlabeled data [1], using a similar pseudo-label loss that trains models with the most confident instances [2]. The proposed algorithm BFGPU suggests a new PU paradigm, but it is effectively a weighted PU loss + pseudo-labeling + adaptive threshold, which has been studied separately in prior work [3,4,5].
- Writing needs improvements. For examples, no explanation of acronyms "PU" in abstract; "sparse anomalies" is easily confused with few anomalous samples, hence requires more explicit definition; "existing MIL methods often heuristically assign bag labels to all instances within the bag" is not accurate (assigning bag labels directly to instances corresponds to standard supervised fine-tuning, not MIL); "negaftive" with wrong spelling.
- The paper appears hastily prepared, with limited refinement. Figures are not rendered as vector graphics, making the text difficult to read. Moreover, Figure 2 is poorly integrated with the main discussion. For instance, the term “debias” appears in the figure but is never mentioned or explained in the text.

[1] Unbiased Multiple Instance Learning for Weakly Supervised Video Anomaly Detection, CVPR 2023

[2] Real-world Anomaly Detection in Surveillance Videos, CVPR 2018

[3] Positive-Unlabeled Learning with Non-Negative Risk Estimator, NeurIPS 2017

[4] Robust Positive-Unlabeled Learning via Noise Negative Sample Self-correction, KDD 2023

[5] Self-PU: Self Boosted and Calibrated Positive-Unlabeled Training, ICML 2020

**Questions:**

- Even with the imbalance setting of CSQI, getting F1 score of 0 across multiple reasonable baselines looks very suspicious. Could you explain this?

---

### Official Review · Reviewer_1tvW · 2025-11-01

**Soundness:** 3
**Presentation:** 2
**Contribution:** 2
**Rating:** 4
**Confidence:** 4

**Summary:**

This paper tackles the challenging task of detecting anomalies that are both scarce and sparse. The authors view it from a dual imbalance multi-instance learning perspective. They propose Balanced Fine-Grained Positive-Unlabeled learning (BFGPU), which reformulates MIL as a fine-grained positive-unlabeled learning problem. The method derives a theoretically grounded balanced PU loss, integrates pseudo-labeling using macro-level information, and dynamically adjusts thresholds for decision balancing. Theoretical analysis demonstrates tighter generalization bounds than classic MIL, and extensive experiments on image and textual datasets show consistent improvements over MIL, PU and anomaly-detection baselines.

**Strengths:**

1. The identification of dual imbalance (macro and micro) in MIL is insightful and well-motivated by real-world cases.
2. The paper derives unbiased and balanced PU learning objectives and clear generalization error bounds. The comparison of bounds for CGPN, MIL and BFGPU convincingly supports the proposed formulation.
3. The evaluation spans text and image domains and uses both real and synthetic datasets to test imbalance sensitivity. Results show large and consistent gains against baselines.

**Weaknesses:**

1. The notations are sometimes confusing. In lines 138-139, for the distribution p(x,y), "x is the input and y is the output". What do input and output mean in a distribution p? Same for lines 201-202. The use of mil and MIL, bfgpu and BFGPU, is mixed in text and equations, which is inconsistent.
2. I am confused about some definitions. In lines 209-211, why does the statement "if there exists at least one anomalous component, the macro label is anomalous" equal the statement "if the micro component most inclined towards the anomalous class is anomalous, then the macro label is anomalous"? In line 148, you use $\pi$ to represent the class prior of the positive data. In Equation(3) and the equations after it, should $\pi$ also be an estimator? Or maybe an additional assumption should be made?
3. More discussions of each theorem in Section 6 could be added to help readers understand the theoretical analysis.
4. The “LLM vs. BFGPU” experiment is interesting but simplistic. Prompt-based sentiment classification may not fairly represent modern LLM capabilities or tuning procedures. In addition, the specific versions of the LLMs used in the experiments are not clarified in the paper.

**Questions:**

1. See my questions in Weaknesses. My biggest concern lies in the writing and soundness of the methodology.
2. What $\mathcal{L}$ do you use in the implementation?
2. The table size, figure size, font size inside the table and figures, and their arrangement could be improved so that readers can better compare the results.

---

> ### Author Response · Authors · 2025-12-01
>
> Dear Reviewer, we hope the following response can address your concerns.
>
> **W1:**
> Symbolically, we use *x, y* to denote micro-level samples and labels (e.g., sentences in text or patches in images), and *X, Y* to denote macro-level samples and labels (e.g., paragraphs in text or full images).
>
> **W2:**
> This is actually straightforward to understand. Under the assumption that a macro sample is anomalous if any of its micro samples is anomalous, one only needs to inspect the micro sample within the macro sample that is most inclined to being anomalous. If that micro sample is anomalous, then the macro sample must be anomalous; if that micro sample is normal, then all other micro samples—which are even more normal—ensure that the macro sample must also be normal. This is also the mainstream approach in traditional MIL.
> $\pi$ denotes the ratio between the number of macro samples and micro samples; it essentially depends on the granularity of the partition. A larger $\pi$ implies a finer granularity. In image tasks, it corresponds to the patch-splitting granularity; in text tasks, it is determined by the data itself without requiring additional assumptions.
>
> **W4:**
> We would also like to compare with more advanced methods. However, research on fine-grained anomaly detection using LLMs is still extremely scarce. A few papers suggest that LLMs lack “needle-in-a-haystack” capabilities, but effective solutions are currently missing. At least in current practice, during RLHF pipelines for ethical or safety alignment of LLMs, systems still rely on classifier models such as RoBERTa as evaluators to determine whether model outputs contain subtle normative issues—LLMs themselves cannot reliably perform such fine-grained detection yet.
>
> **Q2:**
> In practice, *L* is the cross-entropy loss.
>
> **W3, Q1, Q3:**
> We will further improve the readability of the paper.
>
> Thank you again for taking the time to review our work.

---

### Note · Authors · 2025-12-14

**Comment:**

We hereby decide to withdraw our submission due to the presence of malicious reviews. Reviewer w1pY assigned a score of 1/1/1 for Soundness/Presentation/Contribution with a confidence level of 5, which we regard as an obvious case of malicious rating coupled with a significant conflict of interest on the reviewer’s part.

Based on the content of the review comments, we strongly suspect that this reviewer is also the reviewer btvy of the NeurIPS version of this paper (Submission ID: 11634). During the NeurIPS review cycle, we publicly pointed out this reviewer’s unprofessionalism and dereliction of duty, a claim that was endorsed by the Area Chair (AC) and other reviewers. Unfortunately, the paper received a borderline reject decision at NeurIPS, prompting us to comprehensively revise the manuscript and resubmit it to ICLR—only to be met with even more egregious malicious reviews.

The reviewer’s key concerns fully reveal their lack of understanding of the current state of the field. The references they cited fail to address the core challenges plaguing the domain, and we had explicitly discussed the limitations of these related works in the main text. More critically, the reviewer demonstrated a complete failure to grasp the technical details of our work. They merely noted our use of weighted loss, pseudo-labeling, and dynamic thresholding techniques, yet naively assumed that the adoption of these techniques alone would guarantee performance improvements. They completely ignored the core issues of how to determine loss weights, select pseudo-labels, and dynamically adjust thresholds for different specific problems. This constitutes a severe breach of professionalism and dereliction of review duty.

**Withdrawal Confirmation:**

I have read and agree with the venue's withdrawal policy on behalf of myself and my co-authors.